# Beyond Random Masking: When Dropout Meets Graph Convolutional Networks

**Yuankai Luo[1,2] & Xiao-Ming Wu[2] & Hao Zhu[3,*]**
[1]Beihang University, Beijing, China
[2]The Hong Kong Polytechnic University, Hong Kong
[3]Data61♥CSIRO, Sydney, Australia

## Abstract

Graph Convolutional Networks (GCNs) have emerged as powerful tools for learning on graph-structured data, yet the behavior of dropout in these models remains poorly understood. This paper presents a comprehensive theoretical analysis of dropout in GCNs, revealing that its primary role differs fundamentally from standard neural networks - preventing oversmoothing rather than co-adaptation. We demonstrate that dropout in GCNs creates dimension-specific stochastic subgraphs, leading to a form of structural regularization not present in standard neural networks. Our analysis shows that dropout effects are inherently degree-dependent, resulting in adaptive regularization that considers the topological importance of nodes. We provide new insights into dropout's role in mitigating oversmoothing and derive novel generalization bounds that account for graph-specific dropout effects. Furthermore, we analyze the synergistic interaction between dropout and batch normalization in GCNs, uncovering a mechanism that enhances overall regularization. Our theoretical findings are validated through extensive experiments on both node-level and graph-level tasks across 14 datasets. Notably, GCN with dropout and batch normalization outperforms state-of-the-art methods on several benchmarks, demonstrating the practical impact of our theoretical insights.

## 1 Introduction

The remarkable success of deep neural networks across various domains has been accompanied by the persistent challenge of overfitting, where models perform well on training data but fail to generalize to unseen examples. This issue has spurred the development of numerous regularization techniques, among which dropout has emerged as a particularly effective and widely adopted approach (LeCun et al., 2015). Introduced by Srivastava et al. (2014), dropout addresses overfitting by randomly "dropping out" a proportion of neurons during training, effectively creating an ensemble of subnetworks. This technique has proven highly successful in improving generalization and has become a standard tool in the deep learning toolkit. The effectiveness of dropout has prompted extensive theoretical analysis, with various perspectives offered to explain its regularization effects.

Some researchers have interpreted dropout as a form of model averaging (Baldi & Sadowski, 2013), while others have analyzed it through the lens of information theory (Achille & Soatto, 2018). Wager et al. (2013) provided insights into dropout's adaptive regularization properties, and Gal & Ghahramani (2016) established connections between dropout and Bayesian inference. These diverse theoretical frameworks have significantly enhanced our understanding of dropout's role in mitigating overfitting in traditional neural networks. However, as the field of deep learning has expanded to encompass more complex data structures, particularly graphs, new questions have arisen regarding the applicability and behavior of established techniques. Graph Neural Networks (GNNs), especially Graph Convolutional Networks (GCNs), have demonstrated remarkable performance on tasks involving graph-structured data (Kipf & Welling, 2017). Naturally, researchers and practitioners have applied dropout to GNNs, often observing beneficial effects on generalization (Hamilton et al., 2017).

---

*Hao Zhu is the corresponding author and led the writing of the paper.

While dropout was originally designed to prevent co-adaptation of features in standard neural networks, our analysis reveals that its primary mechanism in GCNs is fundamentally different. We demonstrate that dropout's main contribution in GCNs is mitigating oversmoothing by maintaining feature diversity across nodes, rather than preventing co-adaptation as in standard neural networks. This finding represents a significant shift in our understanding of how regularization operates in graph neural networks. Specifically, we demonstrate that:

- Dropout in GCNs creates dimension-specific stochastic sub-graphs, leading to a unique form of structural regularization not present in standard neural networks.

- The effects of dropout are inherently degree-dependent, with differential impacts on nodes based on their connectivity, resulting in adaptive regularization that considers the topological importance of nodes in the graph.

- Dropout plays a crucial role in mitigating the oversmoothing problem rather than co-adaption in GCNs, though its effects are more nuanced than previously thought.

- The generalization bounds for GCNs with dropout exhibit a complex dependence on graph properties, diverging from traditional dropout theory.

- There exists a significant interplay between dropout and batch normalization in GCNs, revealing synergistic effects that enhance the overall regularization.

Our theoretical framework not only provides deeper insights into the mechanics of dropout in graph-structured data but also yields practical implications for the design and training of GCNs. We validate our theoretical findings through extensive experiments on both node-level and graph-level tasks, demonstrating the practical relevance of our analysis. This work bridges a critical gap in the theoretical understanding of regularization in GCNs and paves the way for more principled approaches to leveraging dropout in graph representation learning. Furthermore, we validate our theoretical findings through extensive experiments, demonstrating that GCNs incorporating our insights on dropout and batch normalization outperform several state-of-the-art methods on benchmark datasets. This practical success underscores the importance of our theoretical contributions and their potential to advance the field of graph representation learning.

## 2 RELATED WORK

**Dropout in Neural Networks.** Overfitting can be reduced by using dropout Hinton et al. (2012) to prevent complex co-adaptations on the training data. Since its inception, several variants have been proposed to enhance its effectiveness. DropConnect (Wan et al., 2013) generalizes dropout by randomly dropping connections rather than nodes. Gaussian dropout Srivastava et al. (2014) replaces the Bernoulli distribution with a Gaussian one for smoother regularization. Curriculum dropout (Morerio et al., 2017) adaptively adjusts the dropout rate during training. Theoretical interpretations of dropout have provided insights into its success. The model averaging perspective (Baldi & Sadowski, 2013) views dropout as an efficient way of approximately combining exponentially many different neural networks. The adaptive regularization interpretation (Wager et al., 2013) shows how dropout adjusts the regularization strength for each feature based on its importance. The Bayesian approximation view (Gal & Ghahramani, 2016) connects dropout to variational inference in Bayesian neural networks, providing a probabilistic framework for understanding its effects.

**Regularization in Graph Neural Networks.** Graph Neural Networks (GNNs), while powerful, are prone to overfitting and over-smoothing (Li et al., 2018). Various regularization techniques (Yang et al., 2021; Rong et al., 2020; Fang et al., 2023; Feng et al., 2020) have been proposed to address these issues. DropEdge (Rong et al., 2020) randomly removes edges from the input graph during training, reducing over-smoothing and improving generalization. Graph diffusion-based methods (Gasteiger et al., 2019) incorporate higher-order neighborhood information to enhance model robustness. Spectral-based approaches (Wu et al., 2019) leverage the graph spectrum to design effective regularization strategies. Empirical studies have shown that traditional dropout can be effective in GNNs (Hamilton et al., 2017), but its interaction with graph structure remains poorly understood. Some works have proposed adaptive dropout strategies for GNNs (Gao & Ji, 2019), but these are primarily heuristic approaches without comprehensive theoretical grounding.

**Theoretical Frameworks for GNNs.** Despite the empirical success of Graph Neural Networks (GNNs), establishing theories to explain their behaviors is still an evolving field. Recent works have made significant progress in understanding over-smoothing (Li et al., 2018; Zhao & Akoglu, 2019; Oono & Suzuki, 2019; Rong et al., 2020), interpretability (Ying et al., 2019; Luo et al., 2020; Vu & Thai, 2020; Yuan et al., 2020; 2021), expressiveness (Xu et al., 2018; Chen et al., 2019; Maron et al., 2018; Dehmamy et al., 2019; Feng et al., 2022), and generalization (Scarselli et al., 2018; Du et al., 2019; Verma & Zhang, 2019; Garg et al., 2020; Zhang et al., 2020; Oono & Suzuki, 2019; Lv, 2021; Liao et al., 2020; Esser et al., 2021; Cong et al., 2021). Our work aims to complement these existing theoretical frameworks by focusing on the practical aspects of dropout in GNNs, a widely used regularization technique that has not been thoroughly examined from a theoretical perspective. Previous works have provided valuable insights using classical techniques such as Vapnik-Chervonenkis dimension (Scarselli et al., 2018), Rademacher complexity (Lv, 2021; Garg et al., 2020), and algorithm stability (Verma & Zhang, 2019). Recent efforts (Oono & Suzuki, 2019; Esser et al., 2021) have also made strides in incorporating the transductive learning schema of GNNs into theoretical analyses. We bridge the gap between theoretical understanding and practical implementation of GNNs, offering insights into how dropout affects generalization and performance in graph-structured learning tasks.

## 3 Theoretical Framework

In this section, we develop a rigorous mathematical framework to analyze the behavior of dropout in Graph Convolutional Networks (GCNs). We begin by establishing notations and definitions, then formalize the GCN model with dropout, and finally introduce key concepts that will be central to our analysis.

### 3.1 Notations and Definitions

**Notations.** Let $\mathcal{G} = (\mathcal{V}, \mathcal{E}, \boldsymbol{X})$ be an undirected graph with $n = |\mathcal{V}|$ nodes and $m = |\mathcal{E}|$ edges, where $\mathbf{X} \in \mathbb{R}^{n \times d_0}$ represents the node feature matrix with $d_0$ input features per node. We denote by $\boldsymbol{A} \in \mathbb{R}^{n \times n}$ the adjacency matrix, $\boldsymbol{D} = \mathrm{diag}(deg_1, \ldots, deg_n)$ the degree matrix where $deg_i = \sum_j A_{ij}$, and $\tilde{A} = \boldsymbol{D}^{-\frac{1}{2}} \boldsymbol{A} \boldsymbol{D}^{-\frac{1}{2}}$ the normalized adjacency matrix.

**Graph Convolutional Networks (GCNs).** An L-layer GCN performs the following layer-wise transformation:

$$\boldsymbol{H}^{(l)} = \sigma(\tilde{A} \boldsymbol{H}^{(l-1)} \boldsymbol{W}^{(l)}), \tag{1}$$

where $\boldsymbol{H}^{(l)} \in \mathbb{R}^{n \times d_l}$ is the feature matrix, $\boldsymbol{W}^{(l)} \in \mathbb{R}^{d_{l-1} \times d_l}$ is the weight matrix, $\sigma(\cdot)$ is a non-linear activation, and $\boldsymbol{H}^{(0)} = \boldsymbol{X}$. The feature energy measures representation smoothness:

$$E(\boldsymbol{H}^{(l)}) = \frac{1}{2|\mathcal{E}|} \sum_{(i,j) \in \mathcal{E}} \|\boldsymbol{h}_i^{(l)} - \boldsymbol{h}_j^{(l)}\|_2^2 \tag{2}$$

**Dropout in GCNs.** For layer $l$, dropout applies a random mask $\boldsymbol{M}^{(l)} \in \mathbb{R}^{n \times d_l}$ where each element $M_{ij}^{(l)}$ is drawn independently from Bernoulli$(1 - p)$. The forward pass with dropout is defined as:

$$\boldsymbol{H}^{(l)} = \frac{1}{1 - p} \boldsymbol{M}^{(l)} \odot \sigma(\tilde{A} \boldsymbol{H}^{(l-1)} \boldsymbol{W}^{(l)}), \tag{3}$$

where $\odot$ denotes element-wise multiplication and $p$ is the dropout probability.

**Batch Normalization.** When incorporating batch normalization, the layer transformation becomes:

$$\boldsymbol{H}^{(l)} = \sigma(\mathrm{BN}(\tilde{A} \boldsymbol{H}^{(l-1)} \boldsymbol{W}^{(l)})), \tag{4}$$

where BN applies feature-wise normalization $\mathrm{BN}(\boldsymbol{X}) = \gamma \odot \frac{\boldsymbol{X}_{-\mu_B}}{\sqrt{\sigma_B^2 + \epsilon}} + \beta$ with learnable parameters $\gamma, \beta$ and batch statistics $\mu_B, \sigma_B^2$.

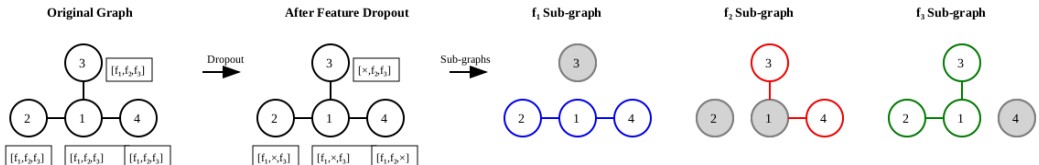

Figure 1: Illustration of how dropout creates dimension-specific sub-graphs. From left to right: the original graph with complete feature vectors, the graph after applying dropout (where $x$ indicates dropped features), and the resulting sub-graphs for each feature dimension. Different colors indicate different feature dimensions, and grayed-out nodes show where features are dropped, preventing message passing along those paths in the next convolution.

## 3.2 DIMENSION-SPECIFIC STOCHASTIC SUB-GRAPHS

We demonstrate how dropout creates dimension-specific sub-graphs in Figure 1. At each iteration $t$, dropout induces **dimension-specific stochastic sub-graphs** $\mathcal{G}_t^{(l,j)} = (\mathcal{V}, \mathcal{E}_t^{(l,j)})$ with:

$$\mathcal{E}_t^{(l,j)} = \{(u,v) \in \mathcal{E} \mid M_{uj}^{(l,t)} \neq 0 \text{ and } M_{vj}^{(l,t)} \neq 0\}. \tag{5}$$

The coupling between feature dropout and graph topology is captured by the feature-topology coupling matrix:

$$C_t^{(l)} = A \odot \mathbb{1}[(M^{(l,t)}(M^{(l,t)})^\top) > 0], \tag{6}$$

which measures how dropout simultaneously affects connected nodes' features. This interaction manifests in each node's effective degree:

$$deg_{i,t}^{\text{eff}} = |\{j \in \mathcal{N}(i) : \exists k, M_{ik}^{(l,t)} \neq 0 \text{ and } M_{jk}^{(l,t)} \neq 0\}| = \sum_j (C_t^{(l)})_{ij}, \tag{7}$$

representing the actual count of node i's neighbors that maintain feature connections after dropout. We consider a path $\mathcal{P} = (v_0, \ldots, v_k)$ active for feature $j$ when all nodes along the path retain this feature, i.e., $\prod_{i=0}^{k-1} M_{v_i j}^{(l,t)} M_{v_{i+1} j}^{(l,t)} \neq 0$. To elucidate the specific impact of dropout on embedding features, we introduce these concepts:

**Theorem 1** (Sub-graph Diversity). *The expected number of distinct sub-graphs per iteration is:*

$$\mathbb{E}[|\mathcal{E}_t^{(l,j)} \mid j = 1, \ldots, d_l|] = d_l(1 - (1 - p)^{2|\mathcal{E}|}),$$

*where $d_l$ is the number of features at layer l, p is the dropout probability, and $|\mathcal{E}|$ is the number of edges in the original graph (The complete proof is in the Appendix. A.1).*

This theorem reveals that dropout in GCNs leads to a rich set of sub-graphs, providing a form of structural data augmentation unique to graph-based models. The diversity of these sub-graphs increases with both the dropout probability $p$ and the number of features $d_l$. This suggests that higher-dimensional GCNs with moderate dropout rates can benefit from a wider range of structural variations during training, potentially leading to more robust and generalizable representations. Moreover, this mechanism allows the GCN to implicitly explore different graph structures without explicitly modifying the input graph. This could be particularly beneficial for tasks where the optimal graph structure is uncertain or where multiple relevant sub-structures exist within the data.

**Theorem 2** (Expected Active Features per Path). *For a path $\mathcal{P}$ of length k, the expected number of features for which it is active is:*

$$\mathbb{E}[\#active\,features\,for\,\mathcal{P}] = d_l(1 - p)^{k+1}.$$

This theorem demonstrates that while individual long paths are unlikely to be active for any given feature, the multi-dimensional nature of GCNs allows for effective long-range information flow through the ensemble effect across features. This theoretical insight is further supported by our empirical analysis in Appendix A.6.

### 3.3 Degree-Dependent Nature of Dropout Effects

The interaction between dropout and the graph structure leads to a form of degree-dependent regularization in GCNs. This means that the effect of dropout varies based on the connectivity of each node, creating an adaptive regularization scheme that considers the topological importance of nodes in the graph.

**Theorem 3** (Degree-Dependent Dropout Effect). *The expected effective degree and its variance are given by:*

$$\mathbb{E}[deg_{i,t}^{eff}] = (1-p)^2 deg_i \ \ and \ \ Var[deg_{i,t}^{eff}] = deg_i(1-p)^2(1-(1-p)^2), \tag{8}$$

*where $deg_i$ is the original degree of node i and p is the dropout probability.*

This theorem highlights that dropout affects nodes differentially depending on their degree. High-degree nodes, typically more influential within the graph, exhibit less variation in their effective degree due to dropout, potentially resulting in more stable representations for these important nodes. This observation is empirically confirmed in the analysis of a 2-layer GCN presented in Appendix A.6. Consequently, the degree-dependent nature of dropout in GCNs results in adaptive regularization, where the regularization effect naturally adjusts to the local graph structure.

**Corollary 4** (Relative Stability of High-Degree Nodes). *The coefficient of variation of the effective degree, defined as $CV[deg_{i,t}^{eff}] = \sqrt{Var[deg_{i,t}^{eff}]}/\mathbb{E}[deg_{i,t}^{eff}]$, decreases with increasing node degree:*

$$CV[deg_{i,t}^{eff}] = \frac{\sqrt{1-(1-p)^2}}{\sqrt{deg_i}(1-p)}.$$

This corollary further confirms that high-degree nodes experience relatively less variation in their effective degree due to dropout. Figure 10 illustrates that the CV decreases as node degree increases. This degree-dependent effect distinguishes dropout in GCNs from its application in standard neural networks and suggests that the optimal dropout strategy for GCNs may need to consider the graph structure explicitly.

### 3.4 Role of Dropout in Oversmoothing

Oversmoothing is a well-known issue in GCNs, where node representations become indistinguishable as the number of layers increases. Our analysis reveals that dropout plays a crucial role in this context, though its effects are more nuanced than previously thought.

**Theorem 5** (Dropout and Feature Energy). *For a GCN with dropout probability p, the expected feature energy at layer l is bounded by:*

$$\mathbb{E}[E(\boldsymbol{H}^{(l)})] \le \frac{deg_{\max}}{|\mathcal{E}|}(\frac{1}{1-p})^l\|\tilde{\boldsymbol{A}}\|_2^{2l}\prod_{i=1}^{l}\|\boldsymbol{W}^{(i)}\|_2^2\|\boldsymbol{X}\|_F^2 \tag{9}$$

*where $E(\boldsymbol{X})$ is the energy of the input features and $\boldsymbol{W}^{(i)}$ are the weight matrices (The complete proof is in the Appendix.A.2).*

The derived bound demonstrates how dropout affects feature energy through the interplay of network depth ($l$), graph structure (through $deg_{\max}$ and $\tilde{\boldsymbol{A}}$), and weight properties ($\|\boldsymbol{W}^{(i)}\|_2^2$). Note that this analysis only provides an upper bound; the absence of a lower bound in this derivation is due to limitations in bounding certain terms. We will later show that when considering batch normalization, we can establish the existence of a lower bound, providing a more complete characterization.

### 3.5 Generalization Bounds with Graph-Specific Dropout Effects

The unique properties of dropout in GCNs, such as the creation of stochastic sub-graphs and degree-dependent effects, influence how these models generalize to unseen data. Our analysis provides novel generalization bounds that explicitly account for these graph-specific dropout effects, offering insights into how dropout interacts with graph structure to influence the model's generalization capabilities.

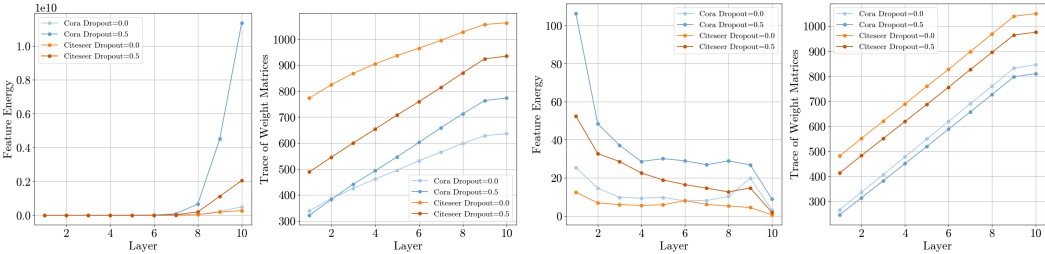

Figure 2: Feature energy vs dropout rates.       Figure 3: BN feature energy vs dropout rates.

**Theorem 6** (Generalization Bound for $L$-Layer GCN with Dropout). *For an L-layer GCN F with dropout probability $p_l$ at layer l and $L_\sigma$-Lipschitz activation function $\sigma$, with probability at least $1 - \delta$ over the training examples, the following generalization bound holds:*

$$\mathbb{E}_D[L(F(X))] - \mathbb{E}_S[L(F(X))] \leq O\left(\sqrt{\frac{\log(1/\delta)}{n}}\right)\sum_{l=1}^{L} L_{loss} \cdot L_l \cdot \sqrt{\frac{p_l}{(1-p_l)\chi_f(\mathcal{G})}}\|\sigma(\tilde{A}H^{(l-1)}W^{(l)})\|_F,$$

(10)

*where $\mathbb{E}_D$ is the expectation over the data distribution, $\mathbb{E}_S$ is the expectation over the training samples, L is the loss function with Lipschitz constant $L_{loss}$, $L_l = \prod_{i=l}^{L}(L_\sigma\|W^{(i)}\|_2 \cdot \|\tilde{A}\|_2)$ is the Lipschitz constant from layer l to output, $\|W^{(i)}\|_2$ is the spectral norm of the weight matrix at layer i, $\|\tilde{A}\|_2$ is the spectral norm of the normalized adjacency matrix, and $\chi_f(\mathcal{G})$ is the fractional chromatic number of the dependency graph $\mathcal{G}$ induced by the message passing structure.*

This generalization bound reveals how the network's stability depends on the loss function's Lipschitz constant, layer-wise Lipschitz constants capturing weight effects, graph structure through $\chi_f(\mathcal{G})$, feature activations, and dropout rates. This leads to several key insights: First, network depth affects stability through the layer-wise Lipschitz constants $L_l$. The multiplicative accumulation of weight and graph effects ($\prod_{i=l+1}^{L}\|W^{(i)}\|\|\tilde{A}\|$) suggests deeper GCNs require careful regularization as perturbations can amplify through layers. Second, the graph structure fundamentally influences stability through $\chi_f(\mathcal{G})$. Since $\chi_f(\mathcal{G}) > 1$ for GCNs due to message passing (versus $\chi_f(\mathcal{G}) = 1$ for MLPs), GCNs gain natural regularization from their graph structure. This effect strengthens with graph connectivity since larger $\chi_f(\mathcal{G})$ leads to better stability. Combined with the fact that the normalized adjacency matrix has bounded spectral norm ($\|\tilde{A}\|_2 \leq 1$), this provides a built-in stabilizing mechanism unique to GNNs. Third, examining layer-specific terms reveals the interplay between weights $\|W^{(l)}\|$, feature magnitudes $\|\sigma(\tilde{A}H^{(l-1)}W^{(l)})\|_F$, and dropout rates $p_l$. The contribution of each layer to the overall bound suggests that adaptive layer-wise dropout rates might be more effective than uniform dropout, particularly when certain layers process more critical features. Finally, the bound mathematically explains the dropout rate trade-off through the term $\sqrt{p_l/((1-p_l)\chi_f(\mathcal{G}))}$. Higher dropout provides stronger regularization but increases noise, while the graph structure (through $\chi_f(\mathcal{G})$) moderates this effect. This helps explain why moderate dropout rates often work best in practice, with the optimal rate depending on the graph's connectivity patterns. This theoretical insight aligns with empirical observations that GNNs often benefit more from dropout than MLPs, as the graph structure provides additional stability through $\chi_f(\mathcal{G})$ while allowing effective information flow via message passing.

### 3.6 Interaction of Dropout and Batch Normalization in GCNs

While dropout provides a powerful regularization mechanism for GCNs, its degree-dependent nature can lead to uneven regularization across nodes. Batch Normalization (BN) offers a complementary approach that can potentially address this issue and enhance the benefits of dropout. Our analysis reveals how the combination of dropout and BN creates a synergistic regularization effect that is sensitive to both graph structure and feature distributions.

**Theorem 7** (Layer-wise Energy Lower Bound for GCN with Dropout and BN). *For an L-layer Graph Convolutional Network with dropout rate p, batch normalization parameters $\{\beta_d^{(l)}, \gamma_d^{(l)}\}_{d=1}^{d_l}$ at*

*each layer l, with probability at least $(1 - \delta)^L$, the expected feature energy at each layer l satisfies:*

$$E(\boldsymbol{H}^{(l)}) \geq \frac{p \cdot deg_{\min}}{2|\mathcal{E}|(1-p)} \sum_{d=1}^{d_l} \Phi(\beta_d^{(l)}/\gamma_d^{(l)}) \cdot (\beta_d^{(l)})^2$$

*where $l = 1, 2, ..., L$ indicates the layer, $deg_{\min}$ is the minimum degree in the graph, $|\mathcal{E}|$ is the total number of edges, $\Phi$ is the standard normal CDF and $\beta_d^{(l)}, \gamma_d^{(l)}$ are the BN parameters for dimension d at layer l (The complete proof is in the Appendix.A.4).*

Our theoretical analysis reveals a crucial interplay between dropout and batch normalization in GCNs. The lower bound on feature energy combines three essential components: (1) A graph structural term $\frac{deg_{min}}{2|\mathcal{E}|}$ that captures the network connectivity, (2) A dropout-induced scaling factor $\frac{p}{1-p}$ that amplifies preserved features, and (3) A BN-controlled feature activation term $\sum_{d=1}^{d_l} \Phi(\beta_d^{(l)}/\gamma_d^{(l)}) \cdot (\beta_d^{(l)})^2$ that establishes a non-zero energy floor. This interaction operates through several key mechanisms: (1) The BN shift parameters $\beta_d^{(l)}$ directly contribute to feature energy through their squared magnitude, while the ratio $\beta_d^{(l)}/\gamma_d^{(l)}$ determines the proportion of features preserved through ReLU activation via the standard normal CDF $\Phi$. Higher positive values of this ratio increase feature preservation. (2) Dropout's $\frac{p}{1-p}$ factor enhances this feature preservation effect, creating a controlled amplification that prevents feature collapse. This amplification is naturally weighted by graph connectivity, with minimum degree $deg_{min}$ ensuring baseline protection even for sparsely connected nodes. (3) The entire bound scales with the graph's minimum degree, illustrating how the mechanism adapts to the underlying graph structure, providing stronger guarantees for more densely connected graphs. This theoretical framework explains our empirical observations in Figures 2 & 3, where batch normalization effectively moderates the energy dynamics in GCNs. By establishing a non-zero lower bound on feature energy, BN prevents complete feature collapse regardless of weight updates, while dropout enhances feature discrimination. Their joint application creates a specialized regularization mechanism for graph-structured data, where BN's parameter-controlled feature preservation interacts with dropout-induced sparsity to maintain robust node representations across graph topologies.

### 3.7 COMPARISON WITH OTHER DROPOUT VARIANTS

Various dropout mechanisms have been proposed for GNNs, each applying masks at different stages of message passing. We formally characterize these variants through their masking operations and their effects in Table 1. The key distinction of standard dropout lies in its feature-dimension-specific masking, which creates unique sub-graph structures for each feature dimension. This leads to a quadratic effect on the effective degree, providing stronger regularization than other variants. While DropNode and DropEdge apply coarse-grained masks uniformly across features, and DropMessage operates at the message level, dropout's feature-specific approach provides finer-grained control over information flow.

Table 1: Comparison of different dropout variants in GNNs. Each method is characterized by its masking operation $M_d$, the resulting sub-graph formation $\mathcal{G}_t$, and expected effective degree $\mathbb{E}[deg_{i,t}^{\text{eff}}]$, where $p$ is the dropout probability.

| Method | Masking Operation | Sub-graph Formation | Expected Effective Degree |
|---|---|---|---|
| DropNode | $M_d = \tilde{\boldsymbol{A}}((M_{node} \odot \boldsymbol{H}^{(l-1)})\boldsymbol{W}^{(l)})_d$ | $\mathcal{G}_t = (\mathcal{V} \setminus \mathcal{V}_{dropped}, \mathcal{E} \setminus \{(i,j)|i \in \mathcal{V}_{dropped}\})$ | $deg_i \prod_{j \in N(i)}(1-p)$ |
| DropEdge | $M_d = (M_{edge} \odot \tilde{\boldsymbol{A}})(\boldsymbol{H}^{(l-1)}\boldsymbol{W}^{(l)})_d$ | $\mathcal{G}_t = (\mathcal{V}, \mathcal{E} \setminus \mathcal{E}_{dropped})$ | $(1-p)deg_i$ |
| DropMessage | $M_d = \tilde{\boldsymbol{A}}(M_{msg_d} \odot (\boldsymbol{H}^{(l-1)}\boldsymbol{W}^{(l)}))_d$ | $\mathcal{G}_t^d = (\mathcal{V}, \{(i,j) \in \mathcal{E}|M_{msg_{d_{ij}}} \neq 0\})$ | $(1-p)deg_i$ |
| Dropout | $M_d = M_{feat_d} \odot \tilde{\boldsymbol{A}}(\boldsymbol{H}^{(l-1)}\boldsymbol{W}^{(l)})_d$ | $\mathcal{G}_t^d = (\mathcal{V}, \{(i,j) \in \mathcal{E}|M_{feat_{d_i}} \neq 0, M_{feat_{d_j}} \neq 0\})$ | $(1-p)^2 deg_i$ |

## 4 EXPERIMENTS

To validate our theoretical analysis, we conducted extensive experiments on a variety of datasets, considering both node-level and graph-level tasks. We implemented dropout technique on several popular GNN architectures: GCN (Kipf & Welling, 2017), GraphSAGE (Hamilton et al., 2017), GAT (Veličković et al., 2018), and GatedGCN (Bresson & Laurent, 2017). For each model, we compared the performance with and without dropout. Our code is available at `https://github.com/LUOyk1999/dropout-theory`.

Table 2: Node classification results (%). The baseline results are taken from Deng et al. (2024); Wu et al. (2023). The top 1st, 2nd and 3rd results are highlighted. "dp" denotes dropout.

| | Cora | CiteSeer | PubMed | Computer | Photo | CS | Physics | WikiCS | ogbn-arxiv | ogbn-products |
|---|---|---|---|---|---|---|---|---|---|---|
| # nodes | 2,708 | 3,327 | 19,717 | 13,752 | 7,650 | 18,333 | 34,493 | 11,701 | 169,343 | 2,449,029 |
| # edges | 5,278 | 4,732 | 44,324 | 245,861 | 119,081 | 81,894 | 247,962 | 216,123 | 1,166,243 | 61,859,140 |
| Metric | Accuracy↑ | Accuracy↑ | Accuracy↑ | Accuracy↑ | Accuracy↑ | Accuracy↑ | Accuracy↑ | Accuracy↑ | Accuracy↑ | Accuracy↑ |
| GCNII | 85.19 ±0.26 | 73.20 ±0.83 | 80.32 ±0.44 | 91.04 ±0.41 | 94.30 ±0.20 | 92.22 ±0.14 | 95.97 ±0.11 | 78.68 ±0.55 | 72.74 ±0.31 | 79.42 ±0.36 |
| GPRGNN | 83.17 ±0.78 | 71.86 ±0.67 | 79.75 ±0.38 | 89.32 ±0.29 | 94.49 ±0.14 | 95.13 ±0.09 | 96.85 ±0.08 | 78.12 ±0.23 | 71.10 ±0.12 | 79.76 ±0.59 |
| APPNP | 83.32 ±0.55 | 71.78 ±0.46 | 80.14 ±0.22 | 90.18 ±0.17 | 94.32 ±0.14 | 94.49 ±0.07 | 96.54 ±0.07 | 78.87 ±0.11 | 72.34 ±0.24 | 78.84 ±0.09 |
| tGNN | 82.97 ±0.68 | 71.74 ±0.49 | 80.67 ±0.34 | 83.40 ±1.33 | 89.92 ±0.72 | 92.85 ±0.48 | 96.24 ±0.24 | 71.49 ±1.05 | 72.88 ±0.26 | 81.79 ±0.54 |
| GraphGPS | 82.84 ±1.03 | 72.73 ±1.23 | 79.94 ±0.26 | 91.19 ±0.54 | 95.06 ±0.13 | 93.93 ±0.12 | 97.12 ±0.19 | 78.66 ±0.49 | 70.97 ±0.41 | OOM |
| NAGphormer | 82.12 ±1.18 | 71.47 ±1.30 | 79.73 ±0.28 | 91.22 ±0.14 | 95.49 ±0.11 | 95.75 ±0.09 | 97.34 ±0.03 | 77.16 ±0.72 | 70.13 ±0.55 | 73.55 ±0.21 |
| Exphormer | 82.77 ±1.38 | 71.63 ±1.19 | 79.46 ±0.35 | 91.47 ±0.17 | 95.35 ±0.22 | 94.93 ±0.01 | 96.89 ±0.09 | 78.54 ±0.49 | 72.44 ±0.28 | OOM |
| GOAT | 83.18 ±1.27 | 71.99 ±1.26 | 79.13 ±0.38 | 90.96 ±0.90 | 92.96 ±1.48 | 94.21 ±0.38 | 96.24 ±0.24 | 77.00 ±0.77 | 72.41 ±0.40 | 82.00 ±0.43 |
| NodeFormer | 82.20 ±0.90 | 72.50 ±1.10 | 79.90 ±1.00 | 86.98 ±0.62 | 93.46 ±0.35 | 95.64 ±0.22 | 96.45 ±0.28 | 74.73 ±0.94 | 59.90 ±0.42 | 73.96 ±0.30 |
| SGFormer | 84.50 ±0.80 | 72.60 ±0.20 | 80.30 ±0.60 | 92.42 ±0.66 | 95.58 ±0.36 | 95.71 ±0.24 | 96.75 ±0.26 | 80.05 ±0.46 | 72.63 ±0.13 | 81.54 ±0.43 |
| Polynormer | 83.25 ±0.93 | 72.31 ±0.78 | 79.24 ±0.43 | 93.68 ±0.21 | 96.46 ±0.26 | 95.53 ±0.16 | 97.27 ±0.08 | 80.10 ±0.67 | 73.46 ±0.16 | 83.82 ±0.11 |
| GCN | 85.22 ±0.66 | 73.24 ±0.63 | 81.08 ±1.16 | 93.15 ±0.34 | 95.03 ±0.24 | 94.41 ±0.13 | 97.07 ±0.04 | 80.14 ±0.52 | 73.13 ±0.27 | 81.87 ±0.41 |
| Dirichlet energy | 74.671 | 9.934 | 4.452 | 8.020 | 3.765 | 20.241 | 8.966 | 6.109 | 8.021 | 7.771 |
| GCN w/o dp | 83.18 ±1.22 | 70.48 ±0.45 | 79.40 ±1.02 | 90.60 ±0.84 | 94.10 ±0.15 | 94.30 ±0.22 | 96.92 ±0.05 | 77.61 ±1.34 | 72.05 ±0.23 | 77.50 ±0.37 |
| Dirichlet energy | 2.951 | 0.170 | 0.247 | 0.592 | 1.793 | 3.980 | 0.318 | 1.592 | 1.231 | 1.745 |
| GCN w/o BN | 84.97 ±0.73 | 72.97 ±0.86 | 80.94 ±0.87 | 92.39 ±0.18 | 94.38 ±0.13 | 93.46 ±0.24 | 96.76 ±0.06 | 79.00 ±0.48 | 71.93 ±0.18 | 79.37 ±0.42 |
| SAGE | 84.14 ±0.63 | 71.62 ±0.29 | 77.86 ±0.79 | 92.65 ±0.21 | 95.71 ±0.20 | 95.90 ±0.09 | 97.20 ±0.10 | 80.29 ±0.97 | 72.72 ±0.13 | 82.69 ±0.28 |
| SAGE w/o dp | 83.06 ±0.80 | 69.68 ±0.82 | 76.40 ±1.48 | 90.17 ±0.60 | 94.90 ±0.17 | 95.80 ±0.08 | 97.06 ±0.06 | 78.84 ±1.17 | 71.37 ±0.31 | 79.82 ±0.22 |
| SAGE w/o BN | 83.89 ±0.67 | 71.39 ±0.75 | 77.26 ±1.02 | 92.54 ±0.24 | 95.51 ±0.23 | 94.87 ±0.15 | 97.03 ±0.03 | 79.50 ±0.93 | 71.52 ±0.17 | 80.91 ±0.35 |
| GAT | 83.92 ±1.29 | 72.00 ±0.91 | 80.48 ±0.99 | 93.47 ±0.27 | 95.33 ±0.16 | 94.49 ±0.17 | 96.73 ±0.10 | 80.21 ±0.68 | 72.83 ±0.19 | 80.05 ±0.34 |
| GAT w/o dp | 82.58 ±1.47 | 71.08 ±0.42 | 79.28 ±0.58 | 92.94 ±0.30 | 93.88 ±0.16 | 94.30 ±0.14 | 96.42 ±0.08 | 78.67 ±0.40 | 71.52 ±0.41 | 77.87 ±0.25 |
| GAT w/o BN | 83.76 ±1.32 | 71.82 ±0.83 | 80.43 ±1.03 | 92.16 ±0.26 | 95.05 ±0.49 | 93.33 ±0.26 | 96.57 ±0.20 | 79.49 ±0.62 | 71.68 ±0.36 | 78.21 ±0.32 |

## 4.1 DATASETS AND SETUP

**Datasets.** For node-level tasks, we used 10 datasets: Cora, CiteSeer, PubMed (Sen et al., 2008), ogbn-arxiv, ogbn-products (Hu et al., 2020), Amazon-Computer, Amazon-Photo, Coauthor-CS, Coauthor-Physics (Shchur et al., 2018), and WikiCS (Mernyei & Cangea, 2020). Cora, CiteSeer, and PubMed are citation networks, evaluated using the semi-supervised setting and data splits from Kipf & Welling (2017). Computer and Photo (Shchur et al., 2018) are co-purchase networks. CS and Physics (Shchur et al., 2018) are co-authorship networks. We used the standard 60%/20%/20% training/validation/test splits and accuracy as the evaluation metric (Chen et al., 2022; Shirzad et al., 2023; Deng et al., 2024). For WikiCS, we adopted the official splits and metrics (Mernyei & Cangea, 2020). For large-scale graphs, we included ogbn-arxiv and ogbn-products with 0.16M to 2.4M nodes, using OGB's standard evaluation settings (Hu et al., 2020).

For graph-level tasks, we used MNIST, CIFAR10 (Dwivedi et al., 2023), and two Peptides datasets (functional and structural) (Dwivedi et al., 2022). MNIST and CIFAR10 are graph versions of their image classification counterparts, constructed using 8-nearest neighbor graphs of SLIC superpixels. We follow all evaluation protocols suggested by Dwivedi et al. (2023). Peptides-func involves classifying graphs into 10 functional classes, while Peptides-struct regresses 11 structural properties. All evaluations followed the protocols in (Dwivedi et al., 2022).

**Baselines.** Our main focus lies on the following prevalent GNNs and transformer models from Polynormer (Deng et al., 2024): GCN (Kipf & Welling, 2017), SAGE (Hamilton et al., 2017), GAT Veličković et al. (2018), GCNII (Chen et al., 2020), (Veličković et al., 2018), APPNP (Gasteiger et al., 2018), GPRGNN (Chien et al., 2020), SGFormer (Wu et al., 2023), Polynormer (Deng et al., 2024), GOAT (Kong et al., 2023), NodeFormer (Wu et al., 2022), NAGphormer (Chen et al., 2022), GTDwivedi & Bresson (2020), SAN Kreuzer et al. (2021), MGT Ngo et al. (2023), DRew Gutteridge et al. (2023), Graph-MLPMixer He et al. (2023), GRIT Ma et al. (2023), GraphGPS (Rampášek et al., 2022), Exphormer (Shirzad et al., 2023), CKGCN (Ma et al., 2024), GRED (Ding et al., 2024), Graph Mamba Behrouz & Hashemi (2024). We report the performance results of baselines primarily from (Deng et al., 2024), with the remaining obtained from their respective original papers or official leaderboards whenever possible, as those results are obtained by well-tuned models.

**Experimental Setup.** We implemented all models using the PyTorch Geometric library (Fey & Lenssen, 2019). The experiments are conducted on a single workstation with 8 RTX 3090 GPUs. For node-level tasks, we adhered to the training protocols specified in (Deng et al., 2024; Luo et al.,

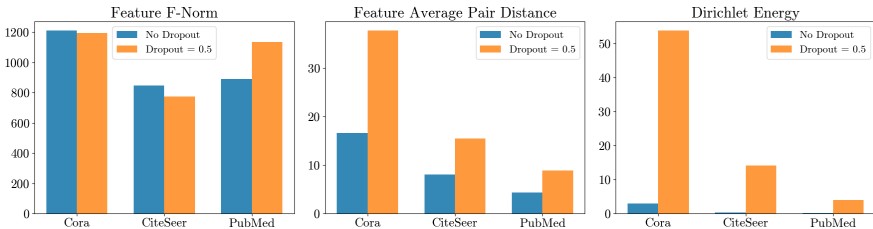

Figure 4: Effect of dropout on feature F-norm, average pair distance, and Dirichlet energy.

2024b;a), employing BN and adjusting the dropout rate between 0.1 and 0.7. In graph-level tasks, we adopted the settings from (Tönshoff et al., 2023; Luo et al., 2025), utilizing BN with a consistent dropout rate of 0.2. All experiments were run with 5 different random seeds, and we report the mean accuracy and standard deviation. To ensure generalizability, we used Dirichlet energy (Cai & Wang, 2020) as an oversmoothing metric, which is proportional to our feature energy.

## 4.2 Node-level Classification Results

The node-level classification results in Table 2 not only align with our theoretical predictions but also showcase the remarkable effectiveness of dropout. Notably, GCN with dropout and batch normalization outperforms state-of-the-art methods on several benchmarks, including Cora, CiteSeer, and PubMed. This superior performance underscores the practical significance of our theoretical insights. Consistently across all datasets, models employing dropout outperform their counterparts without it, validating our analysis that dropout provides beneficial regularization in GNNs, distinct from its effects in standard neural networks. The varying levels of improvement observed across different datasets support our theory of degree-dependent dropout effects that adapt to the graph structure. Furthermore, the consistent increase in Dirichlet energy when using dropout provides empirical evidence for our theoretical insight into dropout's crucial role in mitigating oversmoothing in GCNs, particularly evident in larger graphs. The complementary roles of dropout and batch normalization are demonstrated by the performance drop when either is removed, supporting our analysis of their synergistic interaction in GCNs.

## 4.3 Graph-level Classification Results

Our graph-level classification results, presented in Tables 3 and 4, further validate the broad applicability of our theoretical framework. First, compared to recent SOTA models, we observe that simply tuning dropout enables GNNs to achieve SOTA performance on three datasets and is competitive with the best single-model results on the remaining dataset. Second, the significant accuracy improvements on graph-level tasks such as Peptides-func and CIFAR10 highlight that our insights extend beyond node classification. The varying degrees of improvement across different graph datasets are consistent with our theory that dropout provides adaptive regularization tailored to graph properties. Third, the consistent increase in Dirichlet energy when using dropout supports our theoretical analysis of dropout's role in preserving feature diversity.

These results robustly validate our theory, showing that dropout in GCNs produces dimension-specific stochastic sub-graphs, has degree-dependent effects, mitigates oversmoothing, and offers topology-aware regularization. Combined with batch normalization, dropout enhances GCN performance on graph-level tasks, affirming the relevance and utility of our framework and suggesting directions for improving GNN architectures.

## 4.4 Mitigating Oversmoothing Rather Than Co-adaptation

In traditional neural networks, dropout primarily prevents co-adaptation of neurons. However, our theoretical framework suggests that dropout in GCNs serves a fundamentally different purpose: mitigating oversmoothing rather than preventing co-adaptation. To validate this hypothesis, we examined how dropout affects weight matrices in a 2-layer GCN, focusing specifically on spectral norm changes (see Appendix A.5). We further analyzed three key metrics to quantify dropout's influence on feature representations, as shown in Figure 4. The left panel of Figure 4 demonstrates that the

Table 3: Graph classification results on two pep-Table 4: Graph classification results on two im-
tide datasets from LRGB (Dwivedi et al., 2022). age datasets from (Dwivedi et al., 2023).

| Model | Peptides-func | Peptides-struct |
|---|---|---|
| # graphs | 15,535 | 15,535 |
| Avg. # nodes | 150.9 | 150.9 |
| Avg. # edges | 307.3 | 307.3 |
| Metric | AP ↑ | MAE ↓ |
| GT | $0.6326_{\pm 0.0126}$ | $0.2529_{\pm 0.0016}$ |
| SAN+RWSE | $0.6439_{\pm 0.0075}$ | $0.2545_{\pm 0.0012}$ |
| GraphGPS | $0.6535_{\pm 0.0041}$ | $0.2500_{\pm 0.0012}$ |
| MGT+WavePE | $0.6817_{\pm 0.0064}$ | $0.2453_{\pm 0.0025}$ |
| DRew | $0.7150_{\pm 0.0044}$ | $0.2536_{\pm 0.0015}$ |
| Exphormer | $0.6527_{\pm 0.0043}$ | $0.2481_{\pm 0.0007}$ |
| Graph-MLPMixer | $0.6970_{\pm 0.0080}$ | $0.2475_{\pm 0.0015}$ |
| GRIT | $0.6988_{\pm 0.0082}$ | $0.2460_{\pm 0.0012}$ |
| CKGCN | $0.6952_{\pm 0.0068}$ | $0.2477_{\pm 0.0019}$ |
| GRED | $0.7085_{\pm 0.0027}$ | $0.2503_{\pm 0.0019}$ |
| Graph Mamba | $0.6972_{\pm 0.0100}$ | $0.2477_{\pm 0.0019}$ |
| GCN | $0.7015_{\pm 0.0021}$ | $\mathbf{0.2437}_{\pm 0.0012}$ |
| Dirichlet energy | 9.649 | 6.121 |
| GCN w/o dp | $0.6484_{\pm 0.0034}$ | $0.2541_{\pm 0.0026}$ |
| Dirichlet energy | 6.488 | 3.725 |

| Model | MNIST | CIFAR10 |
|---|---|---|
| # graphs | 70,000 | 60,000 |
| Avg. # nodes | 70.6 | 117.6 |
| Avg. # edges | 564.5 | 941.1 |
| Metric | Accuracy ↑ | Accuracy ↑ |
| GT | $90.831_{\pm 0.161}$ | $59.753_{\pm 0.293}$ |
| SAN+RWSE | - | - |
| GraphGPS | $98.051_{\pm 0.126}$ | $72.298_{\pm 0.356}$ |
| MGT+WavePE | - | - |
| DRew | - | - |
| Exphormer | $98.550_{\pm 0.039}$ | $74.696_{\pm 0.125}$ |
| Graph-MLPMixer | $97.422_{\pm 0.110}$ | $73.961_{\pm 0.330}$ |
| GRIT | $98.108_{\pm 0.111}$ | $76.468_{\pm 0.881}$ |
| CKGCN | $98.423_{\pm 0.155}$ | $72.785_{\pm 0.436}$ |
| GRED | $98.383_{\pm 0.012}$ | $76.853_{\pm 0.185}$ |
| Graph Mamba | $98.392_{\pm 0.183}$ | $74.563_{\pm 0.379}$ |
| GatedGCN | $\mathbf{98.684}_{\pm 0.137}$ | $\mathbf{76.931}_{\pm 0.367}$ |
| Dirichlet energy | 1.119 | 1.541 |
| GatedGCN w/o dp | $98.235_{\pm 0.136}$ | $71.384_{\pm 0.397}$ |
| Dirichlet energy | 0.987 | 0.845 |

Frobenius norm of features remains relatively stable regardless of dropout application, indicating that dropout does not uniformly scale all features. The middle panel reveals that dropout consistently doubles the average pairwise distance between nodes, helping maintain distinct node representations. Most significantly, the right panel shows that dropout substantially increases Dirichlet energy. This dramatic rise in Dirichlet energy, compared to the modest changes in Frobenius norm and pairwise distances, provides compelling evidence that dropout enhances discriminative power between connected nodes, explaining its effectiveness in preventing oversmoothing rather than simply reducing co-adaptation.

## 4.5 COMPARISON WITH DROPOUT VARIANTS

To further explore the practical impact of these different regularization techniques, we conducted hyperparameter tuning for DropEdge, DropNode, and DropMessage on the Cora, Citeseer, and Pubmed datasets. The results, summarized in Table 5, demonstrate that while these methods yield comparable performance, traditional dropout generally performs best.

Table 5: Experimental results of different regularization methods on Cora, Citeseer, and PubMed.

| | Cora (GCN) | CiteSeer (GCN) | PubMed (GCN) | Cora (SAGE) | CiteSeer (SAGE) | PubMed (SAGE) | Cora (GAT) | CiteSeer (GAT) | PubMed (GAT) |
|---|---|---|---|---|---|---|---|---|---|
| GNN | 83.18 ± 1.22 | 70.48 ± 0.45 | 79.40 ± 1.02 | 83.06 ± 0.80 | 69.68 ± 0.82 | 76.40 ± 1.48 | 82.58 ± 1.47 | 71.08 ± 0.42 | 79.28 ± 0.58 |
| GNN+Dropout | **85.22 ± 0.66** | **73.24 ± 0.63** | **81.08 ± 1.16** | **84.14 ± 0.63** | 71.62 ± 0.29 | 77.86 ± 0.79 | **83.92 ± 1.29** | **72.00 ± 0.91** | **80.48 ± 0.99** |
| GNN+DropEdge | 84.88 ± 0.68 | 72.96 ± 0.38 | 80.42 ± 1.15 | 83.10 ± 0.51 | 71.72 ± 0.92 | 77.88 ± 1.31 | 83.44 ± 0.78 | 71.60 ± 1.14 | 79.82 ± 0.68 |
| GNN+DropNode | 84.92 ± 0.52 | 73.08 ± 0.39 | 80.60 ± 0.49 | 83.42 ± 0.58 | **71.92 ± 0.65** | 78.06 ± 1.09 | 83.80 ± 0.97 | 71.30 ± 0.87 | 79.50 ± 0.68 |
| GNN+DropMessage | 84.78 ± 0.58 | 73.12 ± 1.19 | 80.92 ± 0.88 | 83.18 ± 0.62 | 71.22 ± 1.34 | **78.20 ± 0.80** | 83.46 ± 1.06 | 71.38 ± 1.12 | 79.36 ± 1.22 |

## 5 CONCLUSIONS

Our comprehensive theoretical analysis of dropout in GCNs has unveiled complex interactions between regularization, graph structure, and model performance that challenge traditional understanding. These insights not only deepen our understanding of how dropout functions in graph-structured data but also open new avenues for research and development in graph representation learning. Our findings suggest the need to reimagine regularization techniques for graph-based models, explore adaptive and structure-aware dropout strategies, and carefully balance local and global information in GCN architectures. Furthermore, the observed synergies between dropout and batch normalization point towards more holistic approaches to regularization in GNNs. As we move forward, this work lays a foundation for developing more robust and effective graph learning algorithms, with potential applications in dynamic graphs, large-scale graph sampling, and adversarial robustness. Ultimately, this research contributes to bridging the gap between the empirical success of GNNs and their theoretical foundations, paving the way for designing graph learning models.

ACKNOWLEDGMENTS

Hao Zhu was supported by the Science Digital Program in Commonwealth Scientific and Industrial Research Organization (CSIRO). Yuankai Luo received support from National Key R&D Program of China (2021YFB3500700), NSFC Grant 62172026, National Social Science Fund of China 22&ZD153, the Fundamental Research Funds for the Central Universities, State Key Laboratory of Complex & Critical Software Environment (SKLCCSE), and the HK PolyU Grant P0051029.

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

# A    APPENDIX

## A.1    PROOF OF THEOREM 1

*Proof.* Let's approach this proof:

**Step 1:** For a single feature $j$, the probability that an edge is present in the sub-graph $E_t^{(l,j)}$ is $(1-p)^2$, as both endpoints need to retain this feature.

**Step 2:** The probability that an edge is not present in $\mathcal{E}_t^{(l,j)}$ is $1 - (1-p)^2 = p(2-p)$.

**Step 3:** For a sub-graph to be identical to the original graph, all edges must be present. The probability of this is: $((1-p)^2)^{|\mathcal{E}|} = (1-p)^{2|\mathcal{E}|}$.

**Step 4:** Therefore, the probability that $\mathcal{E}_t^{(l,j)}$ is different from the original graph (i.e., unique) is $1 - (1-p)^{2|\mathcal{E}|}$.

**Step 5:** Define an indicator random variable $X_j$ for each feature $j$:

$$X_j = \begin{cases} 1 & \text{if } \mathcal{E}_t^{(l,j)} \text{ is unique} \\ 0 & \text{otherwise} \end{cases}.$$

**Step 6:** We have:

$$P(X_j = 1) = 1 - (1-p)^{2|\mathcal{E}|}][P(X_j = 0) = (1-p)^{2|\mathcal{E}|}.$$

**Step 7:** The expected value of $X_j$ is:

$$\mathbb{E}[X_j] = 1 \cdot P(X_j = 1) + 0 \cdot P(X_j = 0) = 1 - (1-p)^{2|\mathcal{E}|}.$$

**Step 8:** The total number of unique sub-graphs is $\sum_{j=1}^{d_l} X_j$. By the linearity of expectation:

$$\mathbb{E}[|\mathcal{E}_t^{(l,j)} \mid j = 1, \ldots, d_l|] = \mathbb{E}[\sum_{j=1}^{d_l} X_j] = \sum_{j=1}^{d_l} \mathbb{E}[X_j] = d_l(1 - (1-p)^{2|\mathcal{E}|}).$$

This completes the proof.                                                                                        □

### A.2 Proof of Theorem 5

*Proof.* We start with the definition of feature energy:

$$E(\boldsymbol{H}^{(l)}) = \frac{1}{2|\mathcal{E}|} \sum_{i,j \in \mathcal{E}} \|\boldsymbol{h}_i^{(l)} - \boldsymbol{h}_j^{(l)}\|_2^2$$

**Step 1:** Taking the expectation:

$$\mathbb{E}[E(\boldsymbol{H}^{(l)})] = \frac{1}{2|\mathcal{E}|} \sum_{i,j \in \mathcal{E}} \mathbb{E}[\|\boldsymbol{h}_i^{(l)} - \boldsymbol{h}_j^{(l)}\|_2^2].$$

.

**Step 2:** Since $\sum_{(i,j) \in \mathcal{E}}[\|\boldsymbol{h}_i\|^2 + \|\boldsymbol{h}_j\|^2] = 2 \sum_i deg_i \|\boldsymbol{h}_i\|^2$:

$$\frac{1}{2|\mathcal{E}|} \sum_{i,j \in \mathcal{E}} \mathbb{E}[\|\boldsymbol{h}_i^{(l)} - \boldsymbol{h}_j^{(l)}\|_2^2] = \frac{1}{2|\mathcal{E}|} \sum_{i,j \in \mathcal{E}} \mathbb{E}[\|\frac{1}{1-p} \boldsymbol{M}_i^{(l)} \odot \boldsymbol{z}_i^{(l)} - \frac{1}{1-p} \boldsymbol{M}_j^{(l)} \odot \boldsymbol{z}_j^{(l)}\|_2^2]$$

$$= \frac{1}{2|\mathcal{E}|(1-p)^2} \sum_{i,j \in \mathcal{E}} \mathbb{E}[\|\boldsymbol{M}_i^{(l)} \odot \boldsymbol{z}_i^{(l)} - \boldsymbol{M}_j^{(l)} \odot \boldsymbol{z}_j^{(l)}\|_2^2]$$

$$= \frac{1}{2|\mathcal{E}|(1-p)^2} \sum_{i,j \in \mathcal{E}} [(1-p)(\|\boldsymbol{z}_i^{(l)}\|_2^2 + \|\boldsymbol{z}_j^{(l)}\|_2^2) - 2(1-p)^2 (\boldsymbol{z}_i^{(l)})^\top \boldsymbol{z}_j^{(l)}]$$

$$= \frac{1}{1-p} \frac{1}{|\mathcal{E}|} \sum_i deg_i \|\boldsymbol{z}_i^{(l)}\|_2^2 - \frac{1}{|\mathcal{E}|} \mathrm{Tr}(\boldsymbol{Z}^\top \boldsymbol{A} \boldsymbol{Z})$$

where $\boldsymbol{z}_i = \sigma(\sum_k \tilde{\boldsymbol{A}}_{ik} \boldsymbol{h}_k^{(l-1)} \boldsymbol{W}^{(l)})$.

**Step 3:** Since $deg_i \leq deg_{max}$ for all $i$:

$$\frac{1}{|\mathcal{E}|} \sum_i deg_i \|z_i\|_2^2 \leq \frac{deg_{\max}}{|\mathcal{E}|} \sum_i \|z_i\|_2^2 = \frac{deg_{\max}}{|\mathcal{E}|} \|\boldsymbol{Z}\|_F^2.$$

**Step 4:** By ReLU non-negative homogeneity and submultiplicative property:

$$\|\boldsymbol{Z}^{(l)}\|_F^2 \leq \|\tilde{\boldsymbol{A}} \boldsymbol{H}^{(l-1)} \boldsymbol{W}^{(l)}\|_F^2 \leq \|\boldsymbol{W}^{(l)}\|_2^2 \|\tilde{\boldsymbol{A}}\|_2^2 \|\boldsymbol{H}^{(l-1)}\|_F^2$$

**Step 5:** By dropout scaling with probability $p$:

$$\|\boldsymbol{H}^{(l-1)}\|_F^2 = \frac{1}{1-p} \|\boldsymbol{Z}^{(l-1)}\|_F^2$$

**Step 6:** By applying steps 4-5 recursively:

$$\|\boldsymbol{Z}^{(l)}\|_F^2 \leq (\frac{1}{1-p})^{l-1} \|\tilde{\boldsymbol{A}}\|_2^{2l} \prod_{i=1}^{l} \|\boldsymbol{W}^{(i)}\|_2^2 \|\boldsymbol{X}\|_F^2$$

**Step 7:** Combining all inequalities:

$$\mathbb{E}[E(\boldsymbol{H}^{(l)})] \leq \frac{deg_{\max}}{|\mathcal{E}|} (\frac{1}{1-p})^l \|\tilde{\boldsymbol{A}}\|_2^{2l} \prod_{i=1}^{l} \|\boldsymbol{W}^{(i)}\|_2^2 \|\boldsymbol{X}\|_F^2$$

$\square$

### A.3 Proof of Theorem 6

*Proof.* The proof proceeds in several steps:

**Step 1: Dependency Graph.** Let $\mathcal{G} = (\mathcal{V}, \mathcal{E})$ be the dependency graph where vertices $\mathcal{V}$ represent nodes in the graph, and an edge $(i, j) \in \mathcal{E}$ exists if nodes $i$ and $j$ are connected through message passing via $\tilde{\boldsymbol{A}}$. The graph $\mathcal{G}$ is fixed across all layers as it is determined by the structure of $\tilde{\boldsymbol{A}}$.

**Step 2: Dropout Effect as Perturbation.** At layer $l$ with dropout probability $p_l$, let $\delta^{(l)}$ be the perturbation matrix:

$$\delta^{(l)} = \frac{1}{1 - p_l} M^{(l)} \odot \sigma(\tilde{A}H^{(l-1)}W^{(l)}) - \sigma(\tilde{A}H^{(l-1)}W^{(l)}), \tag{11}$$

where $M^{(l)}$ has elements drawn from Bernoulli$(1 - p_l)$.

**Step 3: Perturbation Propagation.** Let $F_l(X)$ denote the network output with dropout applied up to layer $l$. With $L_\sigma$-Lipschitz activation:

$$L_l = \prod_{i=l}^{L} (L_\sigma \|W^{(i)}\|_2 \cdot \|\tilde{A}\|_2) \tag{12}$$

By operator norm properties:

$$\|F_l(X) - F_{l-1}(X)\|_F \le L_l \|\delta^{(l)}\|_F \tag{13}$$

**Step 4: Bounding Matrix Perturbation.** Let $\delta_{ij}^{(l)}$ denote the $(i, j)$-th entry of $\delta^{(l)}$. By Janson's inequality for dependent variables over $\mathcal{G}$ (Zhang & Amini, 2024):

$$\mathbb{E}\left[\sum_{i,j}(\delta_{ij}^{(l)})^2\right] \le \frac{1}{\chi_f(\mathcal{G})} \sum_{i,j} \mathbb{E}[(\delta_{ij}^{(l)})^2] \tag{14}$$

Taking the square root and using the definition of Frobenius norm:

$$\mathbb{E}[\|\delta^{(l)}\|_F] \le \sqrt{\frac{1}{\chi_f(\mathcal{G})} \cdot \mathbb{E}[\|\delta^{(l)}\|_F^2]} \tag{15}$$

$$= \sqrt{\frac{p_l}{(1 - p_l)\chi_f(\mathcal{G})}} \|\sigma(\tilde{A}H^{(l-1)}W^{(l)})\|_F \tag{16}$$

where we use $\mathbb{E}[(M^{(l)})^2] = \mathbb{E}[M^{(l)}] = 1 - p_l$.

**Step 5: Loss Stability.** By the Lipschitz property of the loss function:

$$\mathbb{E}[|L(F_l(x)) - L(F_{l-1}(x))|_F] \le L_{loss} \cdot \mathbb{E}[\|F_l(x) - F_{l-1}(x)\|_F] \tag{17}$$

$$\le L_{loss} \cdot L_l \cdot L_\sigma \cdot \sqrt{\frac{p_l}{(1 - p_l)\chi_f(\mathcal{G})}} \|\sigma(\tilde{A}H^{(l-1)}W^{(l)})\|_F \tag{18}$$

**Step 6: Final Concentration Bound.** Using McDiarmid's inequality and noting the impact of message passing through $\chi_f(\mathcal{G})$, with probability at least $1 - \delta$:

$$\mathbb{E}_D[L(F(x))] - \mathbb{E}_S[L(F(x))] \le O\left(\sqrt{\frac{\log(1/\delta)}{n}}\right) \sum_{l=1}^{L} L_{loss} \cdot L_l \cdot \sqrt{\frac{p_l}{(1 - p_l)\chi_f(\mathcal{G})}} \|\sigma(\tilde{A}H^{(l-1)}W^{(l)})\|_F \tag{19}$$

The bound shows that GNNs ($\chi_f(\mathcal{G}) > 1$ due to message passing) achieve better stability than MLPs ($\chi_f(\mathcal{G}) = 1$, no message passing), with the benefit increasing with graph connectivity. $\square$

### A.4 PROOF OF THEOREM 7

*Proof.* **Step 1:** Start with feature energy and node representation:

$$E(H^{(l)}) = \frac{1}{2|\mathcal{E}|} \sum_{(i,j)\in\mathcal{E}} \|h_i^{(l)} - h_j^{(l)}\|^2$$

$$h_i^{(l)} = \frac{1}{1 - p} M_i^{(l)} \odot z_i^{(l)}$$

where $z_i^{(l)} \in \mathbb{R}^{d_l}$ and $z_i^{(l)} = \sigma(\text{BN}(\sum_k \tilde{A}_{ik} h_k^{(l-1)} W^{(l)}))$

**Step 2:** For the BN output before ReLU at layer $l$, for each feature dimension $d \in \{1, ..., d_l\}$:

$$(\boldsymbol{Y}^{(l)})_{:,d} = \text{BN}((\tilde{\boldsymbol{A}}\boldsymbol{H}^{(l-1)}\boldsymbol{W}^{(l)})_{:,d}) = \gamma_d^{(l)} \frac{(\tilde{\boldsymbol{A}}\boldsymbol{H}^{(l-1)}\boldsymbol{W}^{(l)})_{:,d} - \mu_d^{(l)}}{\sqrt{(\sigma_d^{(l)})^2 + \epsilon}} + \beta_d^{(l)}$$

**Step 3:** For ReLU activation $z = \max(0, y)$ at layer l, for each dimension d:

$$\mathbb{E}[(z_d^{(l)})^2] \geq \Phi(\beta_d^{(l)}/\gamma_d^{(l)}) \cdot (\beta_d^{(l)})^2$$

where $\Phi$ is the standard normal CDF.

**Step 4:** Using the BN-induced bound:

$$\|\boldsymbol{z}_i^{(l)}\|^2 = \sum_{d=1}^{d_l} (z_i^{(l)})_d^2$$

$$\geq \sum_{d=1}^{d_l} \Phi(\beta_d^{(l)}/\gamma_d^{(l)}) \cdot (\beta_d^{(l)})^2 > 0$$

**Step 5:** For feature energy with merged terms:

$$E(\boldsymbol{H}^{(l)}) = \frac{1}{2|\mathcal{E}|} \sum_{(i,j)\in\mathcal{E}} [\frac{1}{1-p}(\|\boldsymbol{z}_i^{(l)}\|^2 + \|\boldsymbol{z}_j^{(l)}\|^2) - 2(\boldsymbol{z}_i^{(l)})^T \boldsymbol{z}_j^{(l)}]$$

$$\geq \frac{1}{2|\mathcal{E}|} \sum_{(i,j)\in\mathcal{E}} [\frac{1}{1-p}(\|\boldsymbol{z}_i^{(l)}\|^2 + \|\boldsymbol{z}_j^{(l)}\|^2) - (\|\boldsymbol{z}_i^{(l)}\|^2 + \|\boldsymbol{z}_j^{(l)}\|^2)]$$

$$= \frac{1}{2|\mathcal{E}|} \sum_{(i,j)\in\mathcal{E}} (\frac{1}{1-p} - 1)(\|\boldsymbol{z}_i^{(l)}\|^2 + \|\boldsymbol{z}_j^{(l)}\|^2)$$

$$= \frac{p}{1-p} \frac{1}{2|\mathcal{E}|} \sum_{(i,j)\in\mathcal{E}} (\|\boldsymbol{z}_i^{(l)}\|^2 + \|\boldsymbol{z}_j^{(l)}\|^2)$$

$$= \frac{p}{1-p} \frac{1}{2|\mathcal{E}|} \sum_i deg_i \|\boldsymbol{z}_i^{(l)}\|^2$$

$$\geq \frac{p \cdot deg_{\min}}{1-p} \frac{1}{2|\mathcal{E}|} \|\boldsymbol{Z}^{(l)}\|_F^2$$

Then with BN bound:

$$E(\boldsymbol{H}^{(l)}) \geq \frac{p \cdot deg_{\min}}{1-p} \frac{1}{2|\mathcal{E}|} \sum_{d=1}^{d_l} \Phi(\beta_d^{(l)}/\gamma_d^{(l)}) \cdot (\beta_d^{(l)})^2$$

$\square$

## A.5 EFFECT OF DROPOUT ON MAX SINGULAR VALUES OF THE WEIGHT MATRICES

We analyze why dropout leads to larger weight matrices in terms of spectral norm $\|\boldsymbol{W}\|_2$. Consider the gradient update for weights $\boldsymbol{W}^{(2)}$ between layers:

$$\frac{\partial L}{\partial \boldsymbol{W}^{(2)}} = (\tilde{\boldsymbol{A}}\boldsymbol{H}_{drop}^{(1)})^\top \times \frac{\partial L}{\partial \boldsymbol{H}^{(2)}} = (\tilde{\boldsymbol{A}}(\boldsymbol{H}^{(1)} \odot \boldsymbol{M}^{(1)})/(1-p))^\top \times \frac{\partial L}{\partial \boldsymbol{H}^{(2)}} \tag{20}$$

where $p$ is the dropout rate and $M^1$ is the dropout mask. This leads to weight updates:

$$\Delta \boldsymbol{W}^{(2)} = -\eta(\tilde{\boldsymbol{A}}\boldsymbol{H}_{drop}^{(1)})^\top \times \frac{\partial L}{\partial \boldsymbol{H}^{(2)}} = -\eta(\tilde{\boldsymbol{A}}(\boldsymbol{H}^{(1)} \odot \boldsymbol{M}^{(1)})/(1-p))^\top \times \frac{\partial L}{\partial \boldsymbol{H}^{(2)}} \tag{21}$$

The $1/(1 - p)$ scaling factor in dropout has two key effects: 1) For surviving features (where $M_{ij}^{(1)} = 1$), the gradient is amplified by $1/(1 - p)$. This leads to larger updates for these weights

during training. 2) During each iteration, different subsets of features survive, but their gradients are consistently scaled up. Over many iterations, this accumulates to larger weight values despite the unbiased expectation maintained by dropout. Specifically, with dropout rate $p$ when $p = 0.5$, surviving gradients are doubled. This amplification effect compounds over training iterations. While dropout maintains unbiased expected values during forward propagation, the consistent gradient scaling during backward propagation leads to systematically larger weight magnitudes. Empirically, we observe that higher dropout rates correlate with larger spectral norms $\|\boldsymbol{W}\|_2^2$ (as shown in Figure 5), supporting this theoretical analysis. The increased weight magnitudes directly contribute to higher feature energy $E(\boldsymbol{H}^{(2)})$ during inference, as:

$$E(\boldsymbol{H}^{(2)}) = \frac{1}{2|\mathcal{E}|} \sum_{(i,j)\in\mathcal{E}} \|\boldsymbol{h}_i^{(2)} - \boldsymbol{h}_j^{(2)}\|_2^2 \tag{22}$$

where larger weights produce more distinctive features between connected nodes, helping mitigate oversmoothing.

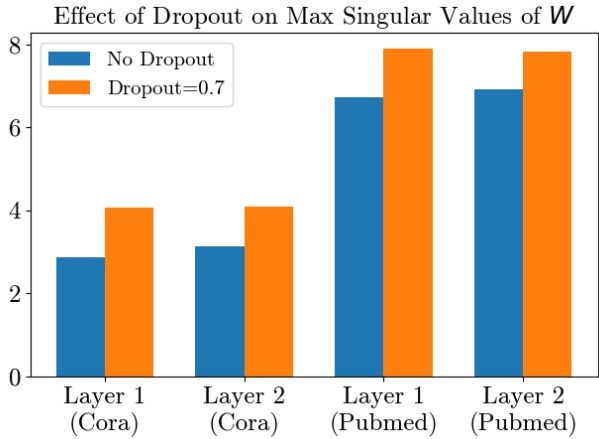

Figure 5: Effect of dropout on max singular values of the weight matrices.

### A.6 Empirical Validation of Theoretical Properties

In this section, we provide empirical evidence supporting the theoretical properties derived in Section 3.

**Dimension-Specific Stochastic Sub-graphs.** Figure 6 shows how varying dropout rates impact the number of edges $\mathcal{E}_t$ in stochastic sub-graphs of a 2-layer GCN, defined by Equation 3, across the Cora and Citeseer datasets. We observe that higher dropout rates correlate with fewer edges in these sub-graphs. This variation demonstrates dropout's role in GCNs as a form of structural regularization, where dimension-specific stochastic sub-graphs are generated. Each feature dimension samples a different sub-graph from the original graph at each iteration. This mechanism provides a rich set of structural variations during training, potentially enhancing the model's ability to capture diverse graph patterns. Figures 7 & 8 illustrate the behavior of active features along paths of length 1 and 2 within a 2-layer GCN equipped with 16 hidden dimensions, across varying dropout rates. Notably, at a dropout rate of 0.6, the average number of active features approaches zero. This characteristic also underscores the importance of multidimensional feature spaces in ensuring robust information transmission under feature dropout.

**Degree-Dependent Nature of Dropout Effects.** Figure 9 demonstrates that dropout affects the effective degree of nodes. Figure 10 illustrates that the CV decreases as node degree increases. This degree-dependent effect distinguishes dropout in GCNs from its application in standard neural networks and suggests that the optimal dropout strategy for GCNs may need to consider the graph structure explicitly. Figure 11 presents empirical evidence supporting Theorem 3 (Degree-Dependent Dropout Effect), which predicts that high-degree nodes experience relatively less variation in their effective degree due to dropout. The figure shows classification accuracy on the Cora

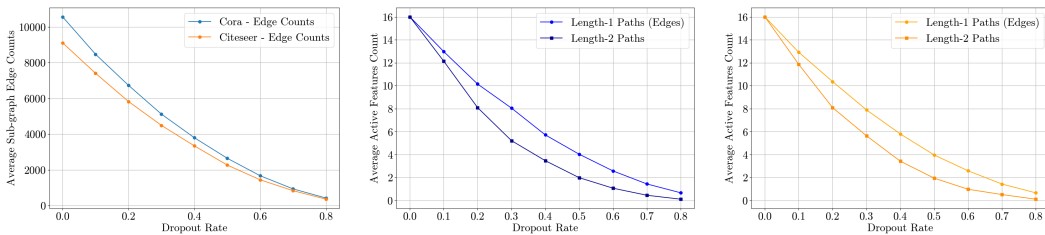

Figure 6: Sub-graph size.   Figure 7: Active path on Cora.  Figure 8: Active path on Citeseer.

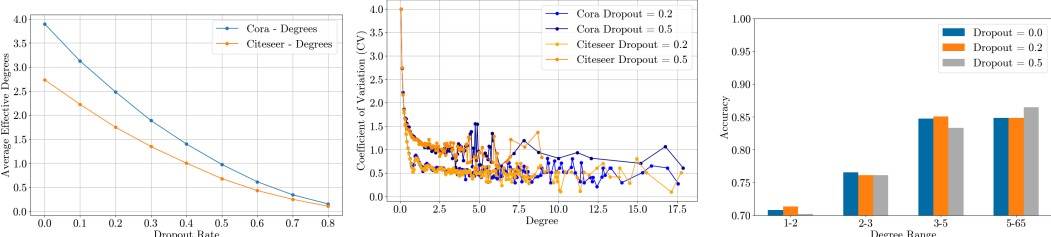

Figure 9: Effective degree.   Figure 10:  Effective CV vs degree.   Figure 11: Accuracy on Cora.

dataset broken down by node degree, demonstrating that nodes with higher degrees consistently achieve better performance. This aligns with our theoretical finding that high-degree nodes maintain more stable representations under dropout, as their effective degree has lower coefficient of variation. The observed pattern confirms that dropout naturally provides adaptive regularization that adjusts to the local graph structure, with stronger stabilizing effects for topologically important nodes.

