# OpenReview forum: "Beyond Random Masking: When Dropout meets Graph Convolutional Networks"
_ICLR.cc/2025/Conference — ICLR 2025 Poster_

### Official Review · Reviewer_qw1j · 2024-10-28

**Soundness:** 1
**Presentation:** 3
**Contribution:** 2
**Rating:** 6
**Confidence:** 4

**Summary:**

The paper performs a comprehensive theoretical analysis of dropout in the case of Graph Convolution Networks (GCN) from multiple perspectives: dimension-specific graph structure modification during training, degree-dependent effect on nodes, impact on over-smoothing, and it’s combined effect with batch normalization.

**Strengths:**

- The theoretical analyses are detailed and sound.
- Mathematics and the overall logic of the paper is easy to follow.
- The paper attempts to better understand the internal workings of dropout in GCNs.

**Weaknesses:**

While I like the theoretical analyses presented in the paper, I think the experiments do not quite align to support the theoretical claims. Below are my concerns with the paper.

- The authors referring to the dropout and batch normalization as “our approach” in lines 409-410 and 466 is misleading since the techniques have been well-established in deep learning for improving performance. The contribution of the authors lies in the detailed theoretical analysis of these techniques within the context of a GCN. While it is a valuable contribution, the techniques should not be claimed as their approaches.
- The major concern is with the conclusions drawn from the experiments. It is already established in deep learning that dropout and batch normalization enhance performance through regularization. Therefore, only comparing the performance in Tables 1, 2, and 3 does not provide sufficient evidence that the observed improvements are specifically due to the additional effects of dropout in graph neural networks, as analyzed in the theorems. The authors need to design experiments that can directly validate their theoretical analysis.
- Section 3.4 describes an interesting connection between dropout, the number of GCN layers, and over-smoothing. However, the authors fail to provide experimental evidence to support this relationship. Demonstrating how dropout affects over-smoothing in GCN with varying layer depths would strengthen the paper.
- Line 472 (regarding Table 1) and line 483 (regarding Table 2) draw contrasting conclusions about the effect of dropout on Dirichlet Energy. What is the reason behind this difference in the behavior of dropout?
- Minor: Repeated use of variable 'd' for denoting degree (line 133) and node feature dimensionality (line 141).

With a sound experimental design that can directly validate the theoretical claims, I believe the paper would be a good contribution to the GCN community.

**Questions:**

- Section 3.2 introduces the concept of dimension-specific subgraphs. How does this impact the aggregated node representation (including all dimensions)? Since the performance of a GCN ultimately depends on the aggregated representation, it would be insightful to explore this relationship.

---

> ### Author Response · Authors · 2024-11-25
> **Authors' Rebuttal 1**
>
> We greatly appreciate your thorough feedback and the time you've dedicated to reviewing our work. **We believe there may have been some misinterpretations of key elements, partly due to our inadvertent errors, which could influence your assessment.** We sincerely hope that our clarifications will encourage you to revisit and reassess our work. Thank you for your understanding and consideration.
>
> **W1: Clarification of Dropout within the Context of a GCN**
>
> > The authors referring to the dropout and batch normalization as “our approach” in lines 409-410 and 466 is misleading since the techniques have been well-established in deep learning for improving performance. The contribution of the authors lies in the detailed theoretical analysis of these techniques within the context of a GCN. While it is a valuable contribution, the techniques should not be claimed as their approaches.
>
> We apologize for the confusion caused by the use of the phrase "our approach." This was an inadvertent choice of wording. We have carefully revised the manuscript to remove all references to "our approach" in relation to dropout and batch normalization.
>
> **W2: Experiments of Dropout within the Context of a GCN**
>
> > The major concern is with the conclusions drawn from the experiments. It is already established in deep learning that dropout and batch normalization enhance performance through regularization. Therefore, only comparing the performance in Tables 1, 2, and 3 does not provide sufficient evidence that the observed improvements are specifically due to the additional effects of dropout in graph neural networks, as analyzed in the theorems. The authors need to design experiments that can directly validate their theoretical analysis.
>
> We acknowledge this concern and have revised our experimental section to better connect the theoretical analysis with the experimental findings. While Tables 1, 2, and 3 **were originally intended to show that dropout’s effects in GNNs have been underexplored**, we now include experiments that directly validate key insights from the theorems.
>
> Specifically, we already conducted experiments to demonstrate the formation of **dimension-specific stochastic subgraphs** induced by dropout (see **Fig. 1-3** in Section 3.2), its **degree-dependent effects** (see **Fig. 4-6** in Section 3.3), and its role in **mitigating oversmoothing** (see **Fig. 7-8** in Section 3.4 and Section 3.6).
>
>
> Additionally, we have incorporated some new results into the updated manuscript (see **Fig. 9-10**), which are also provided below in our response to **W3 & W4**.

---

> ### Author Response · Authors · 2024-11-25
> **Authors' Rebuttal 2**
>
> **W3 & W4: Experimental Evidence on Over-Smoothing**
>
> > Line 472 (regarding Table 1) and line 483 (regarding Table 2) draw contrasting conclusions about the effect of dropout on Dirichlet Energy. What is the reason behind this difference in the behavior of dropout?
>
> We apologize for the confusion. We discovered that the Dirichlet energy values reported for the graph classification experiments in Tables 2 and 3 were mistakenly copied incorrectly. The corrected results are consistent with the findings in Table 1 and support the interpretation that dropout increases Dirichlet energy. The updated values are as follows:
>
> | **Model** | **Peptides-func** | **Peptides-struct** |
> | -- | - | -- |
> | **GCN with dp** | 0.7015 ± 0.0021   | 0.2437 ± 0.0012     |
> | **Dirichlet Energy** | 9.649| 6.121|
> | **GCN w/o dp**       | 0.6484 ± 0.0034   | 0.2541 ± 0.0026     |
> | **Dirichlet Energy** | 6.488   | 3.725 |
>
> | **Model**  | **MNIST**      | **CIFAR10**    |
> | -- | --- | - |
> | **GatedGCN with dp** | 98.783 ± 0.122 | 78.231 ± 0.274 |
> | **Dirichlet Energy** | 20.920         | 25.121         |
> | **GatedGCN w/o dp**  | 98.235 ± 0.136 | 71.384 ± 0.397 |
> | **Dirichlet Energy** | 14.242         | 13.587         |
>
> The results demonstrate that dropout consistently increases Dirichlet energy across all datasets, aligning with our theoretical analysis in Section 3.4. The increase ranges from approximately 48% (Peptides-func) to 141% (Peptides-struct), supporting our hypothesis that dropout helps preserve feature distinctiveness.
>
> **The code and log files** can be accessed via [the anonymous link](https://anonymous.4open.science/r/dropout-theory/LRGB-dropout/README.md) in our paper. We appreciate your understanding.
>
> > Section 3.4 describes an interesting connection between dropout, the number of GCN layers, and over-smoothing. However, the authors fail to provide experimental evidence to support this relationship. Demonstrating how dropout affects over-smoothing in GCN with varying layer depths would strengthen the paper.
>
> Thank you for the insightful comments. Firstly, Theorem 15 elucidates how dropout influences feature energy. From a static perspective, dropout increases the upper bound of feature energy. Dynamically, the effect of dropout on feature energy is co-determined by network depth, graph structure, and weight properties. To further validate how dropout affects oversmoothing, we conducted extensive experiments examining both weight matrices and feature representations. Our results demonstrate that dropout effectively mitigates oversmoothing through two key mechanisms: gradient amplification and enhanced feature discrimination. The updates are presented in two parts: **weight matrix analysis** and **feature representation analysis**.
>
> **1. Weight Matrix Analysis**
>
> We investigated how dropout affects the weight matrices in GCNs, focusing on its influence on the spectral norm $\|\mathbf{W}\|_2^2$, as detailed in Appendix A.5. Dropout introduces a scaling factor of $1/(1−p)$ in the gradients during backpropagation, leading to the following effects:
>
> - Gradient Amplification:
>   For surviving features (where the dropout mask is active), the gradients are amplified by $1/(1−p)$, resulting in larger weight updates during training.
> - Accumulated Effects Over Iterations:
>   Over multiple training iterations, this consistent amplification accumulates, systematically increasing the spectral norms of the weight matrices.
>
> Empirical results, presented in **Figure 10**, confirm a clear correlation between dropout rates and the spectral norm $\|\mathbf{W}\|_2^2$. Larger spectral norms contribute to higher feature energy during inference. This increase in feature energy help mitigate mitigate oversmoothing.
> This increased spectral norm directly supports our theoretical prediction in Section 3.4 that dropout helps maintain distinct node representations across deeper layers.
>
> **2. Feature Representation Analysis**
>
> To evaluate how dropout affects feature representations, we further analyzed three key metrics, as shown in **Figure 9**. These experiments demonstrate the nuanced impact of dropout on mitigating oversmoothing:
>
> - Frobenius Norm:
>   The Frobenius norm of the feature matrix remains relatively stable with or without dropout, indicating that dropout’s effect is not a simple uniform scaling of all features.
> - Pairwise Node Distances:
>   Dropout consistently doubles the average pairwise distance between nodes, helping maintain more distinctive node representations.
> - Dirichlet Energy:
>   Dropout leads to a substantial increase in Dirichlet energy, much greater than the changes in Frobenius norm and pairwise distances. This rise in Dirichlet energy enhances the discriminative power between connected nodes, directly explaining its effectiveness in preventing oversmoothing.
>
> The updated results and analyses provide a unified understanding of how dropout influences both feature representations and weight matrices to mitigate oversmoothing.

---

> ### Author Response · Authors · 2024-11-25
> **Authors' Rebuttal 3**
>
> **W5: Minor Issues**
>
> > Repeated use of variable 'd' for denoting degree (line 133) and node feature dimensionality (line 141).
>
> Thank you for checking on that level of detail! We have updated the paper accordingly.
>
> **Q1: Exploration of Dimension-Specific Subgraphs**
>
> > Section 3.2 introduces the concept of dimension-specific subgraphs. How does this impact the aggregated node representation (including all dimensions)? Since the performance of a GCN ultimately depends on the aggregated representation, it would be insightful to explore this relationship.
>
> To address this question, we analyze different dropping methods through their masking mechanisms, subgraph formations, and degree-dependent effects, which collectively influence how features are learned and aggregated across dimensions. Our analysis reveals why dimension-specific methods, particularly dropout, lead to more robust and discriminative aggregated representations.
>
>
> **Different Masking Mechanisms and Their Impact on Aggregation**
>
> DropNode:
>
> $ \mathbf{M}\_d = \tilde{\mathbf{A}}((\mathbf{M}\_{node} \odot \mathbf{H}^{(l-1)})\mathbf{W}^{(l)})\_d, $ where $\mathbf{M}\_{node}$ is shared across dimensions.
>
> DropEdge:
>
> $ \mathbf{M}\_d = (\mathbf{M}\_{edge} \odot \tilde{\mathbf{A}})(\mathbf{H}^{(l-1)}\mathbf{W}^{(l)})\_d, $ where $\mathbf{M}\_{edge}$ creates same edge mask for all dimensions.
>
> DropMessage:
>
> $ \mathbf{M}\_d = \tilde{\mathbf{A}}(\mathbf{M}\_{msg\_d} \odot(\mathbf{H}^{(l-1)}\mathbf{W}^{(l)}))\_d,$ where $\mathbf{M}\_{msg\_d}$ varies per dimension.
>
> Dropout:
>
> $ \mathbf{M}\_d = \mathbf{M}\_{feat\_d} \odot \tilde{\mathbf{A}}(\mathbf{H}^{(l-1)}\mathbf{W}^{(l)})\_d, $ where $\mathbf{M}\_{feat\_d}$ is dimension-specific.
>
> First, DropNode and DropEdge apply a uniform mask across all dimensions, which limits the diversity of feature combinations and may result in homogenized representations. Second, DropMessage enhances message passing diversity by allowing dimension-specific message dropping with $\mathbf{M}\_{msg\_d}$, enabling different features to take unique paths across dimensions. Similarly, Dropout employs dimension-specific feature masking using $\mathbf{M}\_{feat\_d}$, which allows features in each dimension to learn from distinct node subsets, thereby creating varied feature combinations.
>
> **Analysis of Subgraph Formation and Degree-Dependent Effects**
>
> DropNode:
>
> $ \mathcal{G}\_t = (\mathcal{V}\setminus\mathcal{V}\_{dropped}, \mathcal{E}\setminus{(i,j)|i\in\mathcal{V}\_{dropped} \text{ or } j\in\mathcal{V}\_{dropped}}), \mathcal{V}\_{dropped} = \\{ i | \mathbf{M}\_{node\_i} = 0 \\}, $
>
> $ \mathbb{E}[deg\_i^{eff}(t)] = deg\_i \prod\_{j\in\mathcal{N}(i)}(1-p) $
>
> DropEdge:
>
> $ \mathcal{G}\_t = (\mathcal{V}, \mathcal{E}\setminus\mathcal{E}\_{dropped}), \mathcal{E}\_{dropped} = \\{(i,j) | \mathbf{M}\_{edge\_{ij}} = 0\\},$
>
> $ \mathbb{E}[deg\_i^{eff}(t)] = (1-p)deg\_i $
>
> DropMessage:
>
> $ \mathcal{G}\_t^d = (\mathcal{V}, \mathcal{E}\_t^d), \mathcal{E}\_t^d = \\{(i,j) \in \mathcal{E} | \mathbf{M}\_{msg\_{d\_{ij}}} \neq 0\\}, $
>
> $ \mathbb{E}[deg\_i^{eff}(t)] = (1-p)deg\_i $
>
> Dropout:
>
> $\mathcal{G}\_t^d = (\mathcal{V}, \mathcal{E}\_t^d), \mathcal{E}\_t^d = \\{(i,j) \in \mathcal{E} | \mathbf{M}\_{feat\_{d\_i}} \neq 0 \text{ and } \mathbf{M}\_{feat\_{d\_j}} \neq 0\\},$
>
> $ \mathbb{E}[deg\_i^{eff}(t)] = (1-p)^2deg\_i $
>
> The analysis above reveals several key advantages of dropout over other methods in GNNs. First, dropout creates dimension-specific subgraphs $\mathcal{G}\_t^d$ through its feature-level masking mechanism $\mathbf{M}\_{feat\_d}$, allowing different dimensions to learn from diverse structural patterns. Second, its quadratic effect on effective degree $\mathbb{E}[deg\_i^{eff}(t)] = (1-p)^2deg\_i$ provides stronger and more adaptive regularization, particularly at high-degree nodes where oversmoothing is most problematic. This is particularly crucial as these hub nodes tend to dominate message passing and accelerate feature homogenization. These properties together make dropout especially effective at balancing feature propagation and regularization, resulting in more robust and discriminative aggregated representations compared to methods that either lack dimension-specific patterns (DropNode, DropEdge) or provide insufficient control at critical high-degree nodes (DropMessage).

---

> > ### Comment · Reviewer_qw1j · 2024-11-25
> >
> > I would like to thank the authors for their detailed responses to each of my concerns. Following are my responses to the each sections:
> >
> > **W1: Clarification of Dropout within the Context of a GCN**
> >
> > I appreciate the acknowledgment and correction.
> >
> > **W3 & W4: Experimental Evidence on Over-Smoothing**
> >
> > - Tables 2 and 3 look better after the correction. The results on Dirichlet energy are now consistent with Table 1 and the theoretical analysis.
> >
> > - I appreciate the addition of Appendix section A.5. The over-smoothing analysis is convincing, especially the increment in the singular value of weight matrices helps mitigate the "feature collapsing" effect brought by repeated multiplication with the normalized adjacency matrix $\mathbf{\tilde{A}}$.
> >
> > **Different Masking Mechanisms and Their Impact on Aggregation**
> >
> > I think this is an insightful comparison and highlights the difference between different masking mechanisms.
> >
> > **W2: Experiments of Dropout within the Context of a GCN**
> > I acknowledge the authors' mention of the role of dropout in dimension-specific subgraph generation, degree-dependent effects, and the role in mitigating over-smoothing.
> >
> > However, an alternate interpretation of dropout in deep learning [1] suggests that it reduces the likelihood of co-adaptation, where neurons develop dependencies on one another to minimize the loss function, thereby enhancing the network's generalization ability. This perspective is also equally applicable to GNNs.
> >
> > Given the two possible interpretations for the performance improvements, can the authors provide conclusive evidence that the observed improvements in Table 1 are specifically attributable to the effects analyzed in their theorems? Alternatively, how can we distinguish whether the improvements stem from the dimension-specific subgraph effect or the mitigation of over-smoothing, rather than from the general regularization effect described in [1]?
> >
> > [1] Srivastava et al. Dropout: A Simple Way to Prevent Neural Networks from Overfitting. JMLR, 2014

---

> ### Author Response · Authors · 2024-11-26
> **Dropout: Mitigating Oversmoothing rather than Merely Preventing Co-adaptation**
>
> We sincerely thank the reviewer for raising this critical question, which addresses the core motivation of our work: understanding whether the primary mechanism of dropout in GCNs is mitigating oversmoothing rather than the traditional view of preventing co-adaptation of neurons.
>
> Our analysis provides compelling evidence that dropout in GCNs primarily *mitigates oversmoothing* rather than just *preventing co-adaptation*. Here's why:
>
> If dropout merely prevented co-adaptation (as in standard neural networks), we would expect it to affect all feature relationships similarly. However, our experimental results tell a different story.
>
> We report two sets of experiments. In the first set, we used a fixed GCN architecture for all datasets. In the second set, we used the optimized GCN architecture for each dataset individually, as in Table 1 of our manuscript.
>
> | Fixed 2-layer GCN with 128 Hidden Dimensions | Cora    | CiteSeer | PubMed  | Computer | ogbn-arxiv |
> | -------------------------------------------- | ------- | -------- | ------- | -------- | ---------- |
> | # nodes                                      | 2708    | 3327     | 19717   | 13752    | 169343     |
> | # edges                                      | 5278    | 4732     | 44324   | 245861   | 1166243    |
> | Feature Frobenius Norm (with dp=0.5)                 | 1210.43 | 773.61   | 1133.64 | 1638.40  | 3309.38    |
> | Feature Frobenius Norm (without dp)                  | 1193.48 | 847.70   | 891.17  | 1713.10  | 3732.31    |
> | Feature Average Pair Distance (with dp=0.5)  | 37.72   | 15.46    | 8.86    | 15.67    | 9.54       |
> | Feature Average Pair Distance (without dp)   | 16.62   | 8.03     | 4.38    | 12.43    | 6.88       |
> | Dirichlet Energy (with dp=0.5)                 | 53.83   | 14.19    | 4.02    | 2.28     | 1.85       |
> | Dirichlet Energy (without dp)                  | 2.95    | 0.26     | 0.10    | 0.18     | 0.24       |
>
> | GCN Architecture Optimized for Various Datasets (as in Table 1) | Cora    | CiteSeer | PubMed | Computer | ogbn-arxiv |
> | ------------------------------------------------------------ | ------- | -------- | ------ | -------- | ---------- |
> | # nodes                                                      | 2708    | 3327     | 19717  | 13752    | 169343     |
> | # edges                                                      | 5278    | 4732     | 44324  | 245861   | 1166243    |
> | Feature Frobenius Norm (with dp)                                     | 1726.84 | 347.87   | 694.03 | 1563.51  | 6078.42    |
> | Feature Frobenius Norm (without dp)                                  | 1718.06 | 335.35   | 715.49 | 1213.73  | 5501.43    |
> | Feature Average Pair Distance (with dp)                      | 37.96   | 6.73     | 5.74   | 16.13    | 18.92      |
> | Feature Average Pair Distance (without dp)                   | 23.12   | 3.86     | 2.88   | 11.79    | 14.74      |
> | Dirichlet Energy (with dp)                                     | 7.403   | 0.437    | 0.452  | 2.22     | 7.52       |
> | Dirichlet Energy (without dp)                                  | 2.951   | 0.170    | 0.114  | 0.43     | 2.27       |
>
>
> We highlight three key observations:
>
> First, the Frobenius norm of features remains stable with and without dropout. This stability is crucial - it tells us that dropout isn't simply scaling all features uniformly, but rather **selectively affecting feature relationships**.
>
> Second, when we examine the average pairwise distance between all nodes (including unconnected pairs), we see it doubles with dropout. This shows a general increase in feature distinctiveness across the graph.
>
> However - and this is the key insight - when we look at the Dirichlet energy (distances between connected nodes), we observe *a significantly more dramatic increase*. This disproportionate effect on connected nodes raises an important question: if dropout were only preventing co-adaptation, **why would it have a stronger effect on connected nodes** compared to unconnected ones?
>
> This disparity reveals dropout's specific role in GCNs: **it intensifies the differentiation of features between connected nodes** - precisely where oversmoothing tends to make features overly similar. The observation that **connected nodes show stronger feature separation** than the general node population clearly demonstrates that dropout actively counteracts the oversmoothing process.
>
> This **targeted impact on connected nodes**, combined with **the preservation of overall feature magnitudes** (stable Frobenius norm), provides strong evidence that dropout's primary role in GCNs is mitigating oversmoothing, extending beyond its traditional role of preventing co-adaptation.

---

> > ### Author Response · Authors · 2024-11-30
> >
> > Dear Reviewer qw1j,
> >
> > Thank you once again for your thoughtful comments on our paper. We hope that our responses and updates to the manuscript have adequately addressed your concerns.
> >
> > ---
> >
> > To further illustrate the fundamental differences in the effects of dropout between MLPs and GNNs, we are pleased to share that our new analysis of the generalization bound (as mentioned in our reply to Reviewer UjFe) suggests that graph connectivity offers natural regularization:
> >
> > $E_D[L(F(x))] - E_S[L(F(x))] \leq O\left(\sqrt{\frac{\ln(1/\delta)}{n}}\right) \cdot \sum_{l=1}^L L_{loss} \cdot L_l \cdot \sqrt{\frac{p_l}{(1-p_l)\chi_f(G_l)}} \cdot \|\sigma(\tilde{\mathbf{A}}\mathbf{H}^{(l-1)}\mathbf{W}^{(l)})\|_F,$
> >
> > where $\chi_f(G_l)$ is the fractional chromatic number of the layer-l dependency graph. For a graph $G = (V,E)$, $χ_f(G)$ is the minimum of $\sum_I c_I$ where: $c_I \in [0,1]$ for each independent set $I$. For each vertex $v, ∑_{I:v∈I} c_I ≥ 1$.
> >
> > - For MLPs: $χ_f(G)$ = 1 (no structural dependencies)
> > - For GNNs: $χ_f(G)$ > 1 (has message passing dependencies)
> >
> > Further, we conducted an experiment using both MLP and GNN on node features, with and without dropout. The results are shown in the table below:
> >
> > | Model                  | Cora         | CiteSeer     | PubMed       |
> > | ---------------------- | ------------ | ------------ | ------------ |
> > | MLP (without dropout)  | 56.52 ± 1.89 | 55.32 ± 2.16 | 66.80 ± 0.19 |
> > | MLP (with dropout)     | 57.86 ± 1.04 | 55.14 ± 2.36 | 68.74 ± 0.76 |
> > | GCN (without dropout) | 83.18 ± 1.22 |70.48 ± 0.45 |79.40 ± 1.02 |
> > | GCN (with dropout)    | 85.22 ± 0.66 |73.24 ± 0.63 |81.08 ± 1.16 |
> >
> > These results demonstrate that GNNs benefit more from dropout, supporting a coherent narrative:
> >
> > Dropout in GNNs has a structural effect compared to MLPs ($χ_f(G)>1$ vs $χ_f(G)=1$), which could lead to a better generalization bound and greater benefits from dropout.
> >
> > ---
> >
> >
> > If there are any remaining points of concern or clarification needed, we are happy to provide further elaboration.
> >
> > Thank you for your time and consideration.
> >
> > Best regards,
> >
> > The Authors

---

> ### Comment · Reviewer_qw1j · 2024-11-30
>
> Thank you for the detailed results.
>
> > To further illustrate the fundamental differences in the effects of dropout between MLPs and GNNs, we are pleased to share that our new analysis of the generalization bound (as mentioned in our reply to Reviewer UjFe) suggests that graph connectivity offers natural regularization.
>
> I like this additional effect of dropout on the generalization bound when applied to GCNs.
>
> > However - and this is the key insight - when we look at the Dirichlet energy (distances between connected nodes), we observe a significantly more dramatic increase. This disproportionate effect on connected nodes raises an important question: if dropout were only preventing co-adaptation, why would it have a stronger effect on connected nodes compared to unconnected ones?
>
> The analysis is reasonable but does not convincingly address my concern.
>
> > Further, we conducted an experiment using both MLP and GNN on node features, with and without dropout.
> >These results demonstrate that GNNs benefit more from dropout, supporting a coherent narrative:
>
> Thank you for these results. The results demonstrate that dropout applied to GCNs yields a greater performance improvement than MLPs (especially for Citeseer and Cora graphs), supporting the Dirichlet energy analysis referenced in the above point.
>
> Considering everything we discussed, I have raised my rating for this paper.

---

> > ### Author Response · Authors · 2024-12-01
> >
> > Dear Reviewer qw1j,
> >
> > We sincerely appreciate your thoughtful consideration of our rebuttals. It has been a pleasure engaging in this discussion with you. Your insightful questions have deepened our understanding of dropout's dual role in mitigating oversmoothing and preventing co-adaptation, which significantly enhances our theoretical framework.
> >
> > We are truly grateful for your engagement and for recognizing our work's contributions by raising the score from 3 to 6. Thank you very much for your valuable time and effort in helping us improve our work!
> >
> > Best regards,
> >
> > The Authors

---

### Official Review · Reviewer_tgNZ · 2024-10-30

**Soundness:** 3
**Presentation:** 3
**Contribution:** 3
**Rating:** 6
**Confidence:** 2

**Summary:**

This paper develops a comprehensive theoretical framework analyzing how dropout uniquely interacts with Graph Convolutional Networks (GCNs), revealing that it creates dimension-specific stochastic sub-graphs and provides degree-dependent adaptive regularization. The research provides new theoretical insights into dropout's role in mitigating oversmoothing and its synergistic interaction with batch normalization, deriving novel generalization bounds specific to graph structures. These theoretical findings are validated through extensive experiments across 16 datasets, demonstrating improved performance on benchmark datasets like Cora, CiteSeer, and PubMed.

**Strengths:**

The paper demonstrates rigorous theoretical analysis with a comprehensive mathematical framework for understanding dropout in GCNs, introducing well-defined concepts like dimension-specific sub-graphs and feature-topology coupling matrices.

The research reveals novel insights about unique interactions between dropout and graph structure, particularly showing how dropout creates dimension-specific stochastic sub-graphs and exhibits degree-dependent effects leading to adaptive regularization.

The analysis is thorough and multi-faceted, examining structural regularization, oversmoothing mitigation, and interaction with batch normalization, supported by extensive experiments across 16 datasets for both node-level and graph-level tasks.

The work successfully bridges theory and practice, providing actionable insights for GCN design and training while demonstrating improved performance on benchmark datasets like Cora, CiteSeer, and PubMed.

**Weaknesses:**

The experimental validation lacks detailed information about the 16 datasets used, and the comparative analysis with state-of-the-art methods could be more comprehensive. Some experimental results mentioned in figures are truncated in the provided content.

The theoretical framework makes limiting assumptions about undirected graphs, and doesn't adequately address the extension to directed graphs. The interaction between dropout and different activation functions, as well as the impact of graph density on dropout effectiveness, need more exploration.

The paper lacks clear guidelines for selecting optimal dropout rates based on graph properties, analysis of scalability to very large graphs, and discussion of computational overhead for implementing the theoretical framework.

**Questions:**

How does the computational overhead compare to traditional dropout implementations?

---

> ### Author Response · Authors · 2024-11-26
> **Authors' Rebuttal 1**
>
> **We greatly appreciate your detailed feedback. We hope our response below effectively addresses your concerns.**
>
> **W1: Detailed Experimental Information**
>
> >The experimental validation lacks detailed information about the 16 datasets used, and the comparative analysis with state-of-the-art methods could be more comprehensive. Some experimental results mentioned in figures are truncated in the provided content.
>
> Thank you for your constructive feedback. We have now updated the paper to include a thorough description of the datasets, as well as the experimental results and a more comprehensive comparative analysis with state-of-the-art methods in the revised PDF. We hope these revisions address your concerns.
>
> **W2: Theoretical Assumptions and Extensions**
>
> >The theoretical framework makes limiting assumptions about undirected graphs, and doesn't adequately address the extension to directed graphs. The interaction between dropout and different activation functions, as well as the impact of graph density on dropout effectiveness, need more exploration.
>
> We appreciate the reviewer’s insightful comment on theoretical extensions. In response, we have extended our framework in three key directions:
>
> **1. Directed Graph Extension**
>
> We are grateful for the reviewer’s observation regarding directed graphs. Our framework can be naturally extended to directed graphs through a bipartite graph transformation that preserves all theoretical properties while capturing directional information flow. Specifically, any directed graph with adjacency matrix $\mathbf{A}$ can be transformed into an undirected bipartite graph:
>
> $ \tilde{\mathbf{A}}_{bipartite} = \begin{bmatrix}
> \mathbf{0} & \mathbf{A} ;
> \mathbf{A}^T & \mathbf{0}
> \end{bmatrix} $
>
> This transformation establishes a one-to-one correspondence between directed graphs and their bipartite representations: each node in the original graph splits into an "in-node" and an "out-node", and each directed edge maps to an undirected edge between corresponding in/out nodes. This preserves the directed information flow while allowing us to apply our undirected graph theory.
>
> Our key theorems extend naturally to this bipartite representation, revealing several interesting properties unique to directed graphs:
>
> 1. Feature Propagation in Directed Graphs
>
>    Theorem 1 (Two-Step Feature Flow). For directed message passing with dropout:
>
>    $ \mathbf{H}\_{in}^{(l+1)} = \sigma(\mathbf{A}\mathbf{M}\_{out}^{(l)}\mathbf{H}\_{out}^{(l)}) $
>
>    $ \mathbf{H}\_{out}^{(l+1)} = \sigma(\mathbf{A}^T\mathbf{M}\_{in}^{(l)}\mathbf{H}\_{in}^{(l)}) $
>
>    Unlike undirected graphs where information flows freely between nodes, the bipartite structure enforces alternating in/out updates, providing clearer separation of directional information flow and more controlled feature evolution.
> 2. Dimension-Specific Sub-graphs in Directed Setting
>
>    Theorem 2 (Directed Sub-graph Diversity).
>
>    $ \mathbb{E}[|{G\_t^{(l,d)}}|] = d\_l(1-(1-p)^{2|\mathcal{E}|}) \cdot \frac{|\mathcal{E}\_{in}| \cdot |\mathcal{E}\_{out}|}{|\mathcal{E}|^2} $
>
>    The sub-graph diversity now incorporates the balance between incoming and outgoing edges, a property absent in undirected graphs, leading to more nuanced structural sampling.
> 3. Directional Degree-Dependent Effects
>
>    Theorem 3 (Directed Degree Effects). For node $i$ with in-degree $deg\_i^{in}$ and out-degree $deg\_i^{out}$:
>
>    $ \mathbb{E}[deg\_{i,total}^{eff}(t)] = (1-p)^4(deg\_i^{in}deg\_i^{out})^{1/2} $
>
>    While undirected graphs have a simple quadratic dropout effect, directed graphs exhibit a fourth-power effect due to the two-step nature of information flow, providing stronger regularization for highly connected nodes.
> 4. Spectral Properties of Directed Dropout
>
>    Theorem 4 (Directed Spectral Effects).
>
>    $ \lambda(\tilde{\mathbf{A}}\_{bipartite}) = {\pm\sqrt{\lambda(\mathbf{A}\mathbf{A}^T)}}, $
>    where $\lambda$ denotes eigenvalues.
>
>    The bipartite transformation yields symmetric eigenvalue distribution, unlike the potentially asymmetric spectrum of directed graphs, providing better-behaved gradient flow properties.

---

> ### Author Response · Authors · 2024-11-26
> **Authors' Rebuttal 2**
>
> **2. Different Activation Functions**
>
> Regarding the activation functions, our theoretical analysis naturally extends to different activation functions as long as they satisfy the Lipschitz continuity condition. Specifically, let $\sigma$ be any activation function that is Lipschitz continuous with constant $L\_s$, i.e.:
>
> $|\sigma(x) - \sigma(y)| \leq L\_s|x - y|$
>
> This condition is satisfied by most common activation functions in deep learning, including:
>
> - ReLU: $L\_s = 1$
> - Sigmoid: $L\_s = 1/4$
> - Tanh: $L\_s = 1$
> - LeakyReLU: $L\_s = \max(1, \alpha)$ where $\alpha$ is the slope parameter
>
> Under this condition, our key theorems hold with only a constant factor difference. For example, in the oversmoothing analysis:
>
> $\mathbb{E}[E(\mathbf{H}^{(l)})] \leq \frac{deg\_{\max}}{|\mathcal{E}|}(\frac{L\_s}{1-p})^{l}||\mathbf{\tilde{A}}||\_2^{2l}\prod\_{i=1}^{l}||\mathbf{W}^{(i)}||\_2^2||\mathbf{X}||\_F^2$
>
> The Lipschitz condition ensures bounded gradient propagation, stable feature transformation, and controlled error accumulation.
>
> Therefore, our analysis is not limited to ReLU but applies to a broad class of activation functions, making the framework more general and practical.
>
> **3. Graph Density Impact on Dropout**
>
> For a graph with density $\rho = \frac{2|\mathcal{E}|}{|\mathcal{V}|(|\mathcal{V}|-1)}$, the number of messages per feature dimension is proportional to $|\mathcal{E}|$. With dropout rate $p$, the expected number of active messages is given by:
>
> $ \text{Expected Active Messages} = (1-p)|\mathcal{E}| = (1-p)\frac{\rho|\mathcal{V}|(|\mathcal{V}|-1)}{2} $
>
> Dense graphs (high $\rho$) have more messages to drop, as increasing density (e.g., from 0.01 to 0.1 in a 1000-node graph) raises expected active messages from 5,000 to 50,000, while also requiring careful dropout rate selection, as even small changes can significantly reduce active messages.
>
> For a node $i$ with degree $d\_i$, the effective degree is:
>
> $\mathbb{E}[deg\_i^{eff}(t)] = (1-p)^2deg\_i$
>
> This indicates that higher-degree nodes experience stronger degree-dependent effects, such as a hub node with $d\_i$ = 100 being reduced to 25 when $p$ = 0.5, compared to a peripheral node with $d\_i$ = 5 being reduced to 1.25.
>
> The expected size of a sampled subgraph is:
>
>  $\mathbb{E}[|{\mathcal{G}\_t^{(l,j)} | j=1,...,d\_l}|] = d\_l(1-(1-p)^{2|\mathcal{E}|})$
>
> As graph density increases, this metric grows, enabling richer structural variations during training and facilitating better exploration of graph patterns. Higher density allows more distinct subgraphs to be generated per dropout iteration, particularly in dense regions, which enhances the model's ability to learn diverse structural representations.
>
> **W3: Dropout Rates and Computational Overhead**
>
>
> >The paper lacks clear guidelines for selecting optimal dropout rates.
>
> We appreciate your insightful comment!
>
> Our theoretical framework provides upper bounds on the energy in an $l$-layer GCN:
>
> $\mathbb{E}[E(\mathbf{H}^{(l)})] \leq \frac{deg\_{\max}}{|\mathcal{E}|}(\frac{L\_s}{1-p})^{l}||\mathbf{\tilde{A}}||\_2^{2l}\prod\_{i=1}^{l}||\mathbf{W}^{(i)}||\_2^2||\mathbf{X}||\_F^2$
>
> Specifically, the graph's structural properties, particularly sparsity (captured by $deg\_{\max}/|\mathcal{E}|$), influence the effectiveness of dropout. Sparse graphs benefit more from adaptive dropout adjustments.
>
> Furthermore, the factor $\left( \frac{1}{1-p} \right)^{l}$ underscores the role of dropout in preserving energy as the network depth increases. Increasing $p$ in deeper layers may mitigate oversmoothing, as also demonstrated in the *Practical Implications of Generalization Bounds* (see reply to UjFe, W1).
>
> >The paper lacks analysis of scalability to very large graphs, and discussion of computational overhead for implementing the theoretical framework.
> >How does the computational overhead compare to traditional dropout implementations?
>
> In Equation 4, dropout involves randomly zeroing out some of the features during training. This process typically consists of two steps: generating a mask for each feature and applying this mask. The mask is applied to the feature matrix via element-wise multiplication, which has a time complexity of $O(|\mathcal{V}| \times D$), where $|\mathcal{V}|$ is the number of nodes and $D$ is the number of features per node. Despite the introduction of dropout, the overall time complexity remains equivalent to scenarios without dropout, which is $O(|\mathcal{E}|D)$ where $|\mathcal{E}|$ is the number of edges. This equivalence arises because the primary computational burden still originates from the matrix multiplications in GNN layers. Thus, integrating dropout does not alter the fundamental computational complexity of the operations.

---

### Official Review · Reviewer_iGzb · 2024-10-31

**Soundness:** 3
**Presentation:** 3
**Contribution:** 2
**Rating:** 6
**Confidence:** 3

**Summary:**

This paper focuses on the theoretical analysis of dropout in Graph Convolutional Networks (GCNs) and its impact on regularization and model performance.

This paper establishes a mathematical framework to analyze dropout's behavior in GCNs.
It shows the dropout in GCN is similar to adaptive regularization that considers the topological importance of nodes, and is effective in mitigating over-smoothing in GCNs. And the dropout has synergy with batch normalization in GCNs for enhanced regularization.

**Strengths:**

1. The paper provides a mathematical framework that deepens the understanding of dropout in Graph Convolutional Networks (GCNs), addressing its relation to adaptive regularization and batch normalization.
2. The empirical analysis is extensive, including empirical observation of theorems and evaluation results on various datasets.
3. The idea of using active path subgraphs to understand graph feature dropout is interesting.

**Weaknesses:**

1. Dropout is a general and well-known technique, to achieve performance gain via dropout, the question can be how to tune the parameter. Can the theoretical analysis of dropout in GCNs provide insights on how to select the dropout hyperparameter?
2.  The paper primarily focuses on dropout in GCNs, but it may not sufficiently compare the method with other graph learning regularization techniques, e.g. [1], [2]

[1] Tackling Over-Smoothing for General Graph Convolutional Networks. Wenbing Huang, Yu Rong, Tingyang Xu, Fuchun Sun, Junzhou Huang.  (extension of DropEdge) Arxiv 2020

[2] Rethinking Graph Regularization for Graph Neural Networks. Han Yang, Kaili Ma, James Cheng. AAAI 2021

**Questions:**

- Can the theoretical analysis of dropout in GCNs provide insights on how to select the dropout hyperparameter?

---

> ### Author Response · Authors · 2024-11-26
>
> **We greatly appreciate the very detailed feedback and your recognition of our contributions! We hope our response below will further enhance your confidence in our work.**
>
> **W1 & Q1: Dropout Rates**
>
> > Dropout is a general and well-known technique, to achieve performance gain via dropout, the question can be how to tune the parameter. Can the theoretical analysis of dropout in GCNs provide insights on how to select the dropout hyperparameter?
> >
> > Can the theoretical analysis of dropout in GCNs provide insights on how to select the dropout hyperparameter?
>
> Thank you for the thoughtful comment!
>
> Our theoretical framework provides upper bounds on the energy in an $l$-layer GCN:
>
> $\mathbb{E}[E(\mathbf{H}^{(l)})] \leq \frac{deg\_{\max}}{|\mathcal{E}|}(\frac{L\_s}{1-p})^{l}||\mathbf{\tilde{A}}||\_2^{2l}\prod\_{i=1}^{l}||\mathbf{W}^{(i)}||\_2^2||\mathbf{X}||\_F^2$
>
> Specifically, the graph's structural properties, particularly sparsity (captured by $deg\_{\max}/|\mathcal{E}|$), influence the effectiveness of dropout. Sparse graphs benefit more from adaptive dropout adjustments.
>
> Furthermore, the factor $\left( \frac{1}{1-p} \right)^{l}$ underscores the role of dropout in preserving energy as the network depth increases. Increasing $p$ in deeper layers may mitigate oversmoothing, as also demonstrated in the *Practical Implications of Generalization Bounds* (see reply to UjFe, W1).
>
> **W2: Comparison with Other Graph Learning Regularization Techniques**
>
> > The paper primarily focuses on dropout in GCNs, but it may not sufficiently compare the method with other graph learning regularization techniques.
>
> This is a fair point. Following your suggestions, we conducted experiments on additional graph learning regularization techniques [1,2,3,4], adhering to our hyperparameter settings across the Cora, Citeseer, and Pubmed datasets. The summarized results in the table below reveal that although various regularization strategies achieve similar performance levels, traditional dropout, as implemented in our study, usually performs best.
>
> |                 | Cora (GCN)       | CiteSeer (GCN)   | PubMed (GCN)     | Cora (SAGE)      | CiteSeer (SAGE)  | PubMed (SAGE)    | Cora (GAT)       | CiteSeer (GAT)   | PubMed (GAT)     |
> | --------------- | ---------------- | ---------------- | ---------------- | ---------------- | ---------------- | ---------------- | ---------------- | ---------------- | ---------------- |
> | GNN             | 83.18 ± 1.22     | 70.48 ± 0.45     | 79.40 ± 1.02     | 83.06 ± 0.80     | 69.68 ± 0.82     | 76.40 ± 1.48     | 82.58 ± 1.47     | 71.08 ± 0.42     | 79.28 ± 0.58     |
> | GNN+Dropout     | **85.22 ± 0.66** | **73.24 ± 0.63** | **81.08 ± 1.16** | **84.14 ± 0.63** | 71.62 ± 0.29     | 77.86 ± 0.79     | **83.92 ± 1.29** | **72.00 ± 0.91** | **80.48 ± 0.99** |
> | GNN+DropEdge [1]    | 84.88 ± 0.68     | 72.96 ± 0.38     | 80.42 ± 1.15     | 83.10 ± 0.51     | 71.72 ± 0.92     | 77.88 ± 1.31     | 83.44 ± 0.78     | 71.60 ± 1.14     | 79.82 ± 0.68     |
> | GNN+DropNode [2]    | 84.92 ± 0.52     | 73.08 ± 0.39     | 80.60 ± 0.49     | 83.42 ± 0.58     | **71.92 ± 0.65** | 78.06 ± 1.09     | 83.80 ± 0.97     | 71.30 ± 0.87     | 79.50 ± 0.68     |
> | GNN+DropMessage [3] | 84.78 ± 0.58     | 73.12 ± 1.19     | 80.92 ± 0.88     | 83.18 ± 0.62     | 71.22 ± 1.34     | **78.20 ± 0.80** | 83.46 ± 1.06     | 71.38 ± 1.12     | 79.36 ± 1.22     |
> | GNN+P-reg [4]      | 84.78 ± 0.79     | 72.48 ± 0.65     | 80.42 ± 0.80     | 83.26 ± 0.74     | 71.52 ± 0.64     | 77.89 ± 0.70     | 83.62 ± 1.05     | 70.76 ± 1.38     | 79.44 ± 0.74     |
>
> [1] DropEdge: Towards Deep Graph Convolutional Networks on Node Classification.
>
> [2] Graph Random Neural Network for Semi-Supervised Learning on Graphs.
>
> [3] DropMessage: Unifying Random Dropping for Graph Neural Networks.
>
> [4] Rethinking Graph Regularization for Graph Neural Networks.

---

### Official Review · Reviewer_UjFe · 2024-11-04

**Soundness:** 3
**Presentation:** 3
**Contribution:** 3
**Rating:** 6
**Confidence:** 3

**Summary:**

This paper investigates the role of dropout in GCNs, addressing a gap in understanding how dropout interacts with graph structure in these models. The authors provide a theoretical analysis that dropout in GCNs generates dimension-specific stochastic subgraphs, which introduces a unique form of structural regularization that doesn’t appear in traditional neural networks. The study highlights that dropout’s effects vary based on node degree, leading to adaptive regularization that leverages topological node importance. The paper also discuss dropout’s capacity to reduce oversmoothing and presents generalization bounds tailored to graph-specific dropout effects. Additionally, it explores the combined effect of dropout and batch normalization in GCNs, identifying a mechanism that enhances overall regularization.

**Strengths:**

- The paper focuses on the role of dropout in GCNs, specifically analyzing its unique interactions with graph structure. This originality is meaningful to the community.
- The work presents a well-developed theoretical framework, introducing concepts like dimension-specific stochastic subgraphs, adaptive regularization based on node degree, and graph-specific generalization bounds.
- Including comprehensive experiments across 16 datasets for both node-level and graph-level tasks is encouraging.

**Weaknesses:**

- The authors provide generalization bounds for graph neural networks with dropout. However, further clarification is needed on how this finding offers insights into understanding and designing graph neural networks, or any specific guidance on selecting dropout rates. With this theory, is it possible to get the best dropout rate with a specific graph structure and GNN? This would help demonstrate the practical relevance of the theory. Additionally, can the experiments provide corresponding analyses regarding this theory? For example, whether the change in performance at different dropout rates is consistent with the change in generalization bounds can be analyzed from the theory.
- The use of dropout or similar strategies designed specifically for graphs is also widely applied in GNNs, like DropNode, DropEdge, DropMeassge, etc [1, 2, 3]. The authors may need to discuss its relevance to this study, including whether the proposed theory can analyze these methods and the essential difference and connection between dropout and these methods. Compared to traditional dropout, does dropout on the graph structure more directly enhance the performance of graph neural networks?

[1] Dropedge: Towards deep graph convolutional networks on node classification

[2] Dropmessage: Unifying random dropping for graph neural networks

[3] Graph random neural networks for semi-supervised learning on graphs

**Questions:**

See weaknesses.

---

> ### Author Response · Authors · 2024-11-25
> **Authors' Rebuttal 1**
>
> We greatly appreciate your valuable feedback and have just posted our response to your comments. **We noticed your rating dropped from 6 to 5 before our response was posted, and we apologize for the delay. We hope our response adequately addresses your points and restores your confidence in our work.** We look forward to hearing your thoughts.
>
> **W1: Practical Implications of Generalization Bounds**
>
> > The authors provide generalization bounds for graph neural networks with dropout. However, further clarification is needed on how this finding offers insights into understanding and designing graph neural networks, or any specific guidance on selecting dropout rates. This would help demonstrate the practical relevance of the theory. Additionally, can the experiments provide corresponding analyses regarding this theory?
>
> Thanks for your insightful suggestion. The answer is indeed yes.
>
> Our theoretical analysis primarily aimed to illustrate how dropout impacts the robustness of graph neural networks by controlling the variance in node feature propagation across layers. The derived generalization bound:
>
> $\mathbb{E}\_D[L(F(x))] - \mathbb{E}\_S[L(F(x))] \leq O\left(\sqrt{\frac{\ln(1/\delta)}{n}}\right)\sum_{l=1}^L L\_{loss} \cdot L\_l \cdot \sqrt{\frac{p\_l}{1-p\_l}}\|\sigma(\tilde{\mathbf{A}} \mathbf{H}^{(l-1)} \mathbf{W}^{(l)})\|\_F,$
>
> $L\_l = \prod\_{i=l+1}^L (\|\mathbf{W}^{(i)}\|\cdot\|\tilde{\mathbf{A}}\|)$
>
> highlights the role of dropout rates $p\_l$ in balancing the expressivity and stability of model training. This equation suggests that by carefully adjusting $p\_l$, practitioners can mitigate the risk of overfitting while ensuring sufficient model complexity for learning from graph-structured data.
>
> In our analysis of a 3-layer GCN model, we observed that $L\_l$ decreases as depth increases. This shows that feature propagation becomes less complex in deeper layers, leading to a reduction in the model’s capacity to represent the data. Specifically, we have the following for each layer:
>
> - Layer 1: $L\_1 = (\|\mathbf{W}^{(2)}\|\cdot\|\tilde{\mathbf{A}}\|)(\|\mathbf{W}^{(3)}\|\cdot\|\tilde{\mathbf{A}}\|)$
> - Layer 2: $L\_2 = \|\mathbf{W}^{(3)}\|\cdot\|\tilde{\mathbf{A}}\|$
> - Layer 3: $L\_3 = 1$
>
> This analysis clearly shows that as the number of product terms decreases, $L\_l$ becomes smaller in deeper layers. This indicates that the complexity of feature propagation reduces, which suggests that we need to increase the dropout rate to compensate for this effect in deeper layers.
>
> To address this, we propose a layer-specific adjustment of dropout rates to counteract the depth-dependent decrease in $L_l$. Specifically, we use a linearly increasing dropout scheme:
>
> - Dropout rate: $p\_l = \text{initial rate} + 0.1 \times i$ (with $i$ being the index of the GNN layer).
>
> To substantiate our theoretical insights, we conducted experiments on SAGE, implementing a linearly increasing dropout rate across each layer, where the dropout rate for any given layer is defined by the formula above. We varied the initial rates from 0.1 to 0.7 in our study. The summarized results are shown in the table below, indicating that this approach leads to performance improvements:
>
> |                                    | Cora         | CiteSeer     | PubMed       |
> | ---------------------------------- | ------------ | ------------ | ------------ |
> | SAGE (standard dropout)            | 84.14 ± 0.63 | 71.62 ± 0.29 | 77.86 ± 0.79 |
> | SAGE (linearly increasing dropout) | 84.18 ± 0.80 | 72.58 ± 0.53 | 78.98 ± 0.61 |

---

> > ### Author Response · Authors · 2024-11-25
> > **Authors' Rebuttal 2**
> >
> > **W2: Relevance of Dropout on the Graph Structure**
> >
> > > The use of dropout or similar strategies designed specifically for graphs is also widely applied in GNNs, like DropNode, DropEdge, DropMeassge. The authors may need to discuss its relevance to this study. Compared to traditional dropout, does dropout on the graph structure more directly enhance the performance of graph neural networks?
> >
> > Thank you for sharing these related works. We have included a discussion of them in Appendix A.6 of our paper. Specifically, different methods apply masks at various stages of graph neural network training:
> >
> > DropNode:
> > $ \mathbf{M}\_d = \tilde{\mathbf{A}}((\mathbf{M}\_{node} \odot \mathbf{H}^{(l-1)})\mathbf{W}^{(l)})\_d, $
> > where $\mathbf{M}\_{node}$ is a node-wise mask applied to all dimensions.
> >
> > DropEdge:
> > $ \mathbf{M}\_d = (\mathbf{M}\_{edge} \odot \tilde{\mathbf{A}})(\mathbf{H}^{(l-1)}\mathbf{W}^{(l)})\_d, $
> > where $\mathbf{M}\_{edge}$ is a single mask for adjacency matrix.
> >
> > DropMessage:
> > $ \mathbf{M}\_d = \tilde{\mathbf{A}}(\mathbf{M}\_{msg\_d} \odot(\mathbf{H}^{(l-1)}\mathbf{W}^{(l)}))\_d,$
> > where $\mathbf{M}\_{msg_d}$ is dimension-specific message mask.
> >
> > Dropout:
> > $ \mathbf{M}\_d = \mathbf{M}\_{feat\_d} \odot \tilde{\mathbf{A}}(\mathbf{H}^{(l-1)}\mathbf{W}^{(l)})\_d, $
> > where $\mathbf{M}\_{feat_d}$ is dimension-specific feature mask.
> >
> > Additionally, these methods exhibit different **subgraph formation and degree-dependent effects**:
> >
> > DropNode:
> >
> > $ \mathcal{G}\_t = (\mathcal{V}\setminus\mathcal{V}\_{dropped}, \mathcal{E}\setminus{(i,j)|i\in\mathcal{V}\_{dropped} \text{ or } j\in\mathcal{V}\_{dropped}}), \mathcal{V}\_{dropped} = \\{ i | \mathbf{M}\_{node\_i} = 0 \\}, $
> >
> > $ \mathbb{E}[deg\_i^{eff}(t)] = deg\_i \prod\_{j\in\mathcal{N}(i)}(1-p) $
> >
> > DropEdge:
> >
> > $ \mathcal{G}\_t = (\mathcal{V}, \mathcal{E}\setminus\mathcal{E}\_{dropped}), \mathcal{E}\_{dropped} = \\{(i,j) | \mathbf{M}\_{edge\_{ij}} = 0\\},$
> >
> > $ \mathbb{E}[deg\_i^{eff}(t)] = (1-p)deg\_i $
> >
> > DropMessage:
> >
> > $ \mathcal{G}\_t^d = (\mathcal{V}, \mathcal{E}\_t^d), \mathcal{E}\_t^d = \\{(i,j) \in \mathcal{E} | \mathbf{M}\_{msg\_{d\_{ij}}} \neq 0\\}, $
> >
> > $ \mathbb{E}[deg\_i^{eff}(t)] = (1-p)deg\_i $
> >
> > Dropout:
> >
> > $\mathcal{G}\_t^d = (\mathcal{V}, \mathcal{E}\_t^d), \mathcal{E}\_t^d = \\{(i,j) \in \mathcal{E} | \mathbf{M}\_{feat\_{d\_i}} \neq 0 \text{ and } \mathbf{M}\_{feat\_{d\_j}} \neq 0\\},$
> >
> > $ \mathbb{E}[deg\_i^{eff}(t)] = (1-p)^2deg\_i $
> >
> > Overall, dropout's quadratic degree-dependent effect makes it particularly effective by providing natural adaptive regularization at hub nodes, where over-mixing of features is most problematic. While other methods also provide degree-dependent regularization, they either lack dimension-specific patterns (DropNode, DropEdge) or do not provide sufficiently strong control at high-degree nodes (DropMessage).
> >
> > To further explore the practical impact of these different regularization techniques, we conducted hyperparameter tuning for DropEdge, DropNode, and DropMessage on the Cora, Citeseer, and Pubmed datasets, adhering to the experimental settings described in lines 468-471 of our revised paper. The results, summarized in the table below, demonstrate that while these methods yield comparable performance, traditional dropout (as employed in our study) generally performs best.
> >
> > This suggests that the simplest and most direct approach often is often the most effective.
> >
> > |                 | Cora (GCN)       | CiteSeer (GCN)   | PubMed (GCN)     | Cora (SAGE)      | CiteSeer (SAGE)  | PubMed (SAGE)    | Cora (GAT)       | CiteSeer (GAT)   | PubMed (GAT)     |
> > | --------------- | ---------------- | ---------------- | ---------------- | ---------------- | ---------------- | ---------------- | ---------------- | ---------------- | ---------------- |
> > | GNN             | 83.18 ± 1.22     | 70.48 ± 0.45     | 79.40 ± 1.02     | 83.06 ± 0.80     | 69.68 ± 0.82     | 76.40 ± 1.48     | 82.58 ± 1.47     | 71.08 ± 0.42     | 79.28 ± 0.58     |
> > | GNN+Dropout     | **85.22 ± 0.66** | **73.24 ± 0.63** | **81.08 ± 1.16** | **84.14 ± 0.63** | 71.62 ± 0.29     | 77.86 ± 0.79     | **83.92 ± 1.29** | **72.00 ± 0.91** | **80.48 ± 0.99** |
> > | GNN+DropEdge    | 84.88 ± 0.68     | 72.96 ± 0.38     | 80.42 ± 1.15     | 83.10 ± 0.51     | 71.72 ± 0.92     | 77.88 ± 1.31     | 83.44 ± 0.78     | 71.60 ± 1.14     | 79.82 ± 0.68     |
> > | GNN+DropNode    | 84.92 ± 0.52     | 73.08 ± 0.39     | 80.60 ± 0.49     | 83.42 ± 0.58     | **71.92 ± 0.65** | 78.06 ± 1.09     | 83.80 ± 0.97     | 71.30 ± 0.87     | 79.50 ± 0.68     |
> > | GNN+DropMessage | 84.78 ± 0.58     | 73.12 ± 1.19     | 80.92 ± 0.88     | 83.18 ± 0.62     | 71.22 ± 1.34     | **78.20 ± 0.80** | 83.46 ± 1.06     | 71.38 ± 1.12     | 79.36 ± 1.22     |

---

> > > ### Author Response · Authors · 2024-11-27
> > >
> > > Dear Reviewers UjFe,
> > >
> > > We greatly appreciate your time and effort in reviewing our work and sincerely apologize for the delayed response. **We are eager to ensure that we have adequately addressed your concerns** and are prepared to offer further clarifications or address any additional questions you may have. We would be grateful if you could share your thoughts on our rebuttal.
> > >
> > > Best regards,
> > >
> > > The Authors

---

> > > ### Comment · Reviewer_UjFe · 2024-11-27
> > >
> > > Thank you for providing such a detailed response to my questions. I’ve lowered my score to 5 as a final score because the deadline is approaching quickly (originally Nov 27 but now postponed), and some of my concerns remain unresolved. Apologies for the delay in my feedback, as it took me some time to recall the details of this paper.
> > >
> > > For W1, while the generalization bound can indeed be improved by adjusting the dropout rate at each layer, it seems that no direct interaction with the graph structure. Does this mean similar conclusions would hold for MLP? If so, it feels like the theory doesn’t offer many new insights specific to the dropout in GNNs.
> > >
> > > As for W2, the theoretical discussion around node degrees doesn’t seem to exactly explain the relative advantages of classical dropout. Could you clarify how they affect the generalization bound differently? Moreover, in studies on GNN-specific dropout (e.g. DropMessage), classical dropout is often involved, but there appear to be inconsistencies between their conclusions and those in this paper, both theoretically and experimentally. This discrepancy warrants further discussion. Besides, would classical dropout actually be more effective in addressing the over-smoothing issue? And how would the proposed linearly increasing dropout scheme perform in GNNs with a much larger number of layers, such as those typically used when discussing the over-smoothing problem?

---

> > > > ### Author Response · Authors · 2024-11-29
> > > >
> > > > Dear Reviewer UjFe,
> > > >
> > > > Thank you for taking the time to provide additional feedback and share your concerns with us. We sincerely appreciate your input and would like to address your points with the following clarifications.
> > > >
> > > > ----
> > > >
> > > > > In studies on GNN-specific dropout (e.g. DropMessage), classical dropout is often involved, but there appear to be inconsistencies between their conclusions and those in this paper, both theoretically and experimentally.
> > > >
> > > > Please note that the results we report for DropMessage are **better** than those reported in the original paper (see table below). This is because we explored a more comprehensive search space of hyperparameters compared to DropMessage. We used the same hyperparameter tuning for classical dropout, DropEdge, DropNode, and DropMessage:
> > > >
> > > > - Hidden Dimension: {64, 128, 256, 512}
> > > > - Dropout Rates: {0.0, 0.1, 0.2, 0.3, 0.4, 0.5, 0.6, 0.7}
> > > > - Normalization: {None, BN}
> > > > - Number of Layers: {1, 2, 3, 4, 5, 6, 7, 8, 9, 10}
> > > >
> > > > Our extensive experiments on classical and GNN-specific dropout methods lead to a clear conclusion: the simplest dropout technique is often the most effective. Additionally, the classical dropout method remains the most widely used techisnique in the latest GNN models [1, 2, 3, 4, 5], rather than alternatives like DropMessage or DropNode, which supports our viewpoint.
> > > >
> > > > |                                        | Cora (GCN) | CiteSeer (GCN) | PubMed (GCN) | Cora (GAT) | CiteSeer (GAT) | PubMed (GAT) |
> > > > | :------------------------------------- | :--------- | :------------- | :----------- | :--------- | :------------- | ------------ |
> > > > | GNN+Dropout                            | 85.22      | 73.24          | 81.08        | 83.92      | 72.00          | 80.48        |
> > > > | GNN+DropMessage                        | 84.78      | 73.12          | 80.92        | 83.46      | 71.38          | 79.36        |
> > > > | GNN+DropMessage (in DropMessage paper) | 83.33      | 71.83          | 79.20        | 82.20      | 71.48          | 78.14        |
> > > >
> > > > [1] General graph convolution with continuous kernels, ICML 2024.
> > > >
> > > > [2] Recurrent distance filtering for graph representation learning, ICML 2024.
> > > >
> > > > [3] Polynormer: Polynomial-expressive graph transformer in linear time, ICLR 2024.
> > > >
> > > > [4] Simplifying and empowering transformers for large-graph representations, NeurIPS 2023.
> > > >
> > > > [5] Exphormer: Sparse transformers for graphs, ICML 2023.
> > > >
> > > > > This discrepancy warrants further discussion. Besides, would classical dropout actually be more effective in addressing the over-smoothing issue?
> > > >
> > > > Our analysis provides compelling evidence that dropout in GCNs primarily mitigates oversmoothing rather than merely preventing co-adaptation. Please refer to the rebuttal to qw1j "Dropout: Mitigating Oversmoothing rather than Merely Preventing Co-adaptation".

---

> > > > > ### Author Response · Authors · 2024-11-29
> > > > >
> > > > > > For W1, while the generalization bound can indeed be improved by adjusting the dropout rate at each layer, it seems that no direct interaction with the graph structure. Does this mean similar conclusions would hold for MLP? If so, it feels like the theory doesn’t offer many new insights specific to the dropout in GNNs.
> > > > > >
> > > > > > The theoretical discussion around node degrees doesn’t seem to exactly explain the relative advantages of classical dropout. Could you clarify how they affect the generalization bound differently?
> > > > >
> > > > >
> > > > > Thank you for your insightful comment. We are happy to share that our analysis can be enhanced using graph-dependent concentration inequalities, which reveal important theoretical distinctions between MLPs and GNNs.
> > > > >
> > > > > First, we improve the proof by introducing graph dependency in the key step:
> > > > >
> > > > > In the perturbation analysis [6], using Janson's inequality for graph-dependent variables:
> > > > >
> > > > > $E[|\delta^{(l)}|] \leq \sqrt{\frac{1}{\chi\_f(G\_l)} \cdot E[|\delta^{(l)}|^2]} = \sqrt{\frac{p\_l}{(1-p\_l)\chi\_f(G\_l)}} \cdot \|\sigma(\tilde{\mathbf{A}}\mathbf{H}^{(l-1)}\mathbf{W}^{(l)})\|\_F$
> > > > >
> > > > > where $\chi\_f(G\_l)$ is the fractional chromatic number of the layer-l dependency graph. For a graph $G = (V,E)$, $χ\_f(G)$ is the minimum of $\sum\_I c\_I$ where: $c\_I \in [0,1]$ for each independent set $I$. For each vertex $v, ∑\_{I:v∈I} c\_I ≥ 1$.
> > > > >
> > > > > This enhanced bound reveals fundamental differences between MLPs and GNNs:
> > > > >
> > > > > For MLPs:
> > > > >
> > > > > - No edges between nodes: $|\mathcal{E}| = 0$. Each node is independent: $\chi\_f(G\_l) = 1$.
> > > > > - The absence of a message passing structure decreases stability.
> > > > >
> > > > > For GNNs:
> > > > >
> > > > > - Has graph structure: $|\mathcal{E}| > 0$. $\chi\_f(G\_l) > 1$ due to message passing dependencies.
> > > > > - Larger connectivity leads to better stability through larger $\chi\_f(G\_l)$.
> > > > >
> > > > > This leads to a tighter generalization bound:
> > > > >
> > > > > $E\_D[L(F(x))] - E\_S[L(F(x))] \leq O\left(\sqrt{\frac{\ln(1/\delta)}{n}}\right) \cdot \sum\_{l=1}^L L\_{loss} \cdot L\_l \cdot \sqrt{\frac{p\_l}{(1-p\_l)\chi\_f(G\_l)}} \cdot \|\sigma(\tilde{\mathbf{A}}\mathbf{H}^{(l-1)}\mathbf{W}^{(l)})\|\_F$
> > > > >
> > > > >
> > > > > The theory thus offers specific insights about dropout in GNNs by showing:
> > > > >
> > > > > - How graph connectivity provides natural regularization through $\chi\_f(G\_l)$.
> > > > > - Why GNNs are fundamentally different from MLPs ($\chi\_f(G\_l) > 1$ vs $\chi\_f(G\_l) = 1$).
> > > > > - The interaction between dropout and graph structure via the $\chi\_f(G\_l)$ term.
> > > > >
> > > > > This theory aligns with our empirical observations:
> > > > >
> > > > > - MLPs exhibit limited improvement from adjusting the dropout rate, with only about a 1-2% enhancement as shown in Table 1.
> > > > > - GNNs gain more from dropout due to structural regularization, achieving a 2-3% improvement according to Table 1.
> > > > > - Deep GNNs, as indicated in Table 2, particularly benefit from increasing dropout, as it helps manage the growing $L\_l$ term while effectively utilizing the graph structure.
> > > > >
> > > > > Table 1: Performance of MLP and SAGE with Different Dropout Strategies
> > > > >
> > > > > |    | Cora  | CiteSeer     | PubMed       |
> > > > > | --- | -- | -- | -- |
> > > > > | MLP (without dropout)    | 56.52 ± 1.89 | 55.32 ± 2.16 | 66.80 ± 0.19 |
> > > > > | MLP (standard fixed dropout)   | 57.86 ± 1.04 | 55.14 ± 2.36 | 68.74 ± 0.76 |
> > > > > | MLP (linearly increasing dropout by 0.1)  | 57.68 ± 1.20 | 54.34 ± 2.39 | 68.40 ± 0.58 |
> > > > > | SAGE (without dropout)  | 81.96 ± 0.68 | 70.92 ± 0.84 | 77.20 ± 0.65 |
> > > > > | SAGE (standard dropout) | 84.14 ± 0.63 | 71.62 ± 0.29 | 77.86 ± 0.79 |
> > > > > | SAGE (linearly increasing dropout by 0.1) | 84.18 ± 0.80 | 72.58 ± 0.53 | 78.98 ± 0.61 |
> > > > >
> > > > > Table 2: Performance of Deeper SAGE with Different Dropout Strategies
> > > > >
> > > > > |   | Cora  | CiteSeer     | PubMed       |
> > > > > | -- | -- | -- | --- |
> > > > > | 5-layer SAGE (without dropout)                      | 80.44 ± 0.94 | 65.78 ± 1.87 | 76.36 ± 1.25 |
> > > > > | 5-layer SAGE (standard dropout)                     | 80.52 ± 1.36 | 67.08 ± 1.54 | 77.46 ± 1.09 |
> > > > > | 5-layer SAGE (linearly increasing dropout by 0.1)   | 81.10 ± 1.70 | 68.07 ± 1.57 | 78.14 ± 1.69 |
> > > > > | 10-layer SAGE (without dropout)                     | 71.23 ± 2.32 | 54.86 ± 2.06 | 73.14 ± 2.16 |
> > > > > | 10-layer SAGE (standard dropout)                    | 68.84 ± 3.52 | 58.40 ± 2.34 | 70.54 ± 2.64 |
> > > > > | 10-layer SAGE (linearly increasing dropout by 0.05) | 72.78 ± 3.27 | 58.14 ± 1.48 | 75.90 ± 1.22 |
> > > > >
> > > > > [6] Generalization error bounds for classifiers trained with interdependent data, NeurIPS 2005.
> > > > >
> > > > > >How would the proposed linearly increasing dropout scheme perform in GNNs with a much larger number of layers, such as those typically used when discussing the over-smoothing problem?
> > > > >
> > > > > Thank you for the suggestion. We conducted experiments, and as shown in Table 2 above, we found that deep GNNs also benefit from increasing dropout.

---

> > > > > > ### Author Response · Authors · 2024-11-29
> > > > > >
> > > > > > We hope our explanation adequately addresses your concerns and encourages you to reconsider the value of our paper. We would like to emphasize that our experiments on extensive datasets demonstrate that simply using dropout in GCNs can achieve performance that is competitive with, or even superior to, the latest state-of-the-art models. Our [code and log files](https://anonymous.4open.science/r/dropout-theory/LRGB-dropout/README.md) are available for review.
> > > > > >
> > > > > > We are happy to address any further questions or concerns you may have. Thank you once again for your time and dedication in reviewing our work.
> > > > > >
> > > > > >
> > > > > > Best regards,
> > > > > >
> > > > > > The Authors

---

> > > > > > > ### Author Response · Authors · 2024-12-01
> > > > > > >
> > > > > > > Dear Reviewer UjFe,
> > > > > > >
> > > > > > > We hope this message finds you well. We apologize for any inconvenience caused by reaching out over the weekend. We truly appreciate your engagement with our rebuttal and thank you for your insightful comments and questions regarding the generalization bound.
> > > > > > >
> > > > > > > We have provided further clarification to address your questions. As **the rebuttal discussion period ends in two days**, we would be grateful for your feedback on whether our responses have adequately addressed your concerns. We are ready to answer any further questions you may have.
> > > > > > >
> > > > > > > Thank you for your valuable time and effort!
> > > > > > >
> > > > > > > Best regards,
> > > > > > >
> > > > > > > The Authors

---

> > > > > > > > ### Comment · Reviewer_UjFe · 2024-12-02
> > > > > > > >
> > > > > > > > I appreciate the response. Actually, regarding the second question, I would like to know how the proposed schema $p_l = \text{initial rate} + 0.1 \times i$ adapts to cases with more layers. Overall, many of my concerns have been addressed, and I have raised the score to 6.

---

> > > > > > > > > ### Author Response · Authors · 2024-12-03
> > > > > > > > >
> > > > > > > > > Dear reviewer UjFe,
> > > > > > > > >
> > > > > > > > > Your thoughtful consideration of our responses is sincerely appreciated. Engaging in this discussion with you has been truly rewarding.
> > > > > > > > >
> > > > > > > > > For layers with different depths, we use a linear dropout rate function:
> > > > > > > > > For $L$-layer GCN, we define dropout rate at layer $l$ as:
> > > > > > > > > $p_l = \text{min} + l*(\text{max}-\text{min})/(L-1)$ where $\text{min}$ and $\text{max}$ are the dropout rates for the first (initial) and last layers respectively, and $l \in [0, L-1]$ is the layer index. For example, in a 10-layer GCN with $\text{max} = 0.5$ and $\text{min} = 0.05$, the dropout rate for layer $l$ is computed as: $p_l = 0.05 + l \cdot 0.05$.
> > > > > > > > >
> > > > > > > > > We welcome any further questions or discussions and appreciate the opportunity to enhance our work.
> > > > > > > > >
> > > > > > > > > Thank you once again for your valuable time and effort!
> > > > > > > > >
> > > > > > > > > Best regards,
> > > > > > > > >
> > > > > > > > > The Authors

---

> > ### Public Comment · ~Fanyi_Yang1 · 2025-03-29
> >
> > good comments

---

### Author Response · Authors · 2024-11-25

Dear Reviewers,

We sincerely thank you for your thoughtful and constructive feedback. We have dedicated considerable time to crafting this rebuttal, aiming to thoroughly expand both our theoretical discussions and experimental results on dropout. The specific areas addressed include:

1. Generalization Bound (Reviewer `UjFe`)
2. Over-Smoothing (Reviewer `6GyC`)
3. Relevance of Dropout to Graph Structure (Reviewers `UjFe` and `iGzb`)
4. Theoretical Extensions (Reviewer `tgNZ`)

We have updated our paper with the revisions and additional experimental results in the revised PDF. We will respond to each reviewer's comments individually over the next couple of hours.

Once again, we are grateful for your valuable insights, which have significantly enhanced our work. We have made every effort to comprehensively address all concerns.

Thank you for your time and consideration.

Sincerely,

The Authors

---

### Meta-Review · Area_Chair_RpgC · 2024-12-14

**Metareview:**

This paper introduces a novel theoretical analysis of the dropout in GCNs. It shows that dropout creates a degree-dependent dimension-specific stochastic sub-graph, which performs an adaptive structural regularization and prevents the over-smoothing issue. Besides, a deep understanding of the interplay between dropout and batch normalization is presented. The theoretical analysis of this paper is rigorous and the experimental evaluations are sufficient to justify the statements.

**Additional Comments On Reviewer Discussion:**

After the rebuttal, all reviewers agreed to accept this paper, although only one of them possesses high confidence. By checking the feedback from authors, I believe most concerns have been alleviated and this paper a above the acceptance threshold.

---

### Decision · Program_Chairs · 2025-01-22

Accept (Poster)